# Balancing selection on genomic deletion polymorphisms in humans

Alber Aqil[1], Leo Speidel[2,3], Pavlos Pavlidis[4], Omer Gokcumen[1]*

[1]Department of Biological Sciences, University at Buffalo, Buffalo, United States; [2]University College London, Genetics Institute, London, United Kingdom; [3]The Francis Crick Institute, London, United Kingdom; [4]Institute of Computer Science (ICS), Foundation of Research and Technology-Hellas, Heraklion, Greece

**Abstract:** A key question in biology is why genomic variation persists in a population for extended periods. Recent studies have identified examples of genomic deletions that have remained polymorphic in the human lineage for hundreds of millennia, ostensibly owing to balancing selection. Nevertheless, genome-wide investigation of ancient and possibly adaptive deletions remains an imperative exercise. Here, we demonstrate an excess of polymorphisms in present-day humans that predate the modern human-Neanderthal split (ancient polymorphisms), which cannot be explained solely by selectively neutral scenarios. We analyze the adaptive mechanisms that underlie this excess in deletion polymorphisms. Using a previously published measure of balancing selection, we show that this excess of ancient deletions is largely owing to balancing selection. Based on the absence of signatures of overdominance, we conclude that it is a rare mode of balancing selection among ancient deletions. Instead, more complex scenarios involving spatially and temporally variable selective pressures are likely more common mechanisms. Our results suggest that balancing selection resulted in ancient deletions harboring disproportionately more exonic variants with GWAS (genome-wide association studies) associations. We further found that ancient deletions are significantly enriched for traits related to metabolism and immunity. As a by-product of our analysis, we show that deletions are, on average, more deleterious than single nucleotide variants. We can now argue that not only is a vast majority of common variants shared among human populations, but a considerable portion of biologically relevant variants has been segregating among our ancestors for hundreds of thousands, if not millions, of years.

*For correspondence:
gokcumen@gmail.com

**Competing interest:** The authors declare that no competing interests exist.

## Editor's evaluation

Detecting and quantifying balancing selection is a notoriously difficult challenge. In this study, the authors use both empirical analyses and simulations to characterize the amount of balancing selection in the human genome, focusing specifically on the contribution of polymorphic deletions. These results will be of interest to population and human geneticists.

## Introduction

The evolutionary forces that shape the allele frequency distribution of functional genetic variants remain a hotly debated issue. In humans, tens of thousands of common variants are reported to be associated with human diseases (*Loos, 2020*). However, the mainstream view remains that the majority of these functional genetic variants have had a negligible effect on reproductive fitness and that the frequency of these variants has fluctuated neutrally by drift over time (*Bromberg et al., 2013*; *Dudley et al., 2012*). Functional variants that have measurable fitness effects are often observed at a low frequency (*Eyre-Walker, 2010*). These low-frequency functional variants are considered to be

**eLife digest** The persistence of versions of genes that cause severe disease in human populations has long perplexed scientists. It is common for many versions of a gene to exist. But scientists expect that over time natural selection will eliminate versions of genes harmful to human health.

Sometimes, there are good reasons that a disease-causing gene may persist. For example, having two copies of a particular gene variant causes a condition, called sickle cell disease. But having one sickle cell-causing copy of the gene and one non-disease-causing copy protects against malaria. As a result, the version of the gene that causes sickle cell is more common in people from areas where malaria is prevalent despite the risks to people who end up with two copies. Scientists call this phenomenon balancing selection because trade-offs in the gene's benefits and risks cause it to persist in the population.

Aqil et al. show that balancing selection has likely caused many ancient gene variants to persist in human populations. In the experiments, Aqil et al. scoured the genomes of hundreds of modern humans from around the world and four groups of ancient human ancestors, including Neanderthals and Denisovans. The experiments looked for structural changes in genes, like deletions, that date back to more than 700,000 years ago – before modern humans split from their ancestors. They found large numbers of such ancient genes in modern humans.

Using computer modeling, Aqil et al. showed that these ancient genes likely persist because of balancing selection. Many of these ancient genes regulate the immune response and metabolism. These genes may protect against infectious diseases outbreaks and starvation, which have occurred periodically throughout human history. But these same genes may cause immune or metabolic diseases in modern humans not currently facing these threats. The experiments show how such biological trade-offs have shaped human evolution and reveal that modern human populations, regardless of race or region of origin, share the same genetic variation that already our ancestors carried within them.

in the process of being eliminated from the population by negative selection (*Gibson, 2018*; *Lettre, 2014*; *Zeng et al., 2018*). Nevertheless, an increasing number of studies are showing that more complex evolutionary histories (*Benton et al., 2021*; *Mathieson and Mathieson, 2018*) involving introgression from archaic hominins (*McArthur et al., 2020*), geography-specific adaptation (*Hamid et al., 2021*; *Lachance and Tishkoff, 2013*; *Mendoza-Revilla et al., 2021*), negative selection (*Zeng et al., 2018*), and polygenic selection (*Barghi et al., 2020*; *Berg and Coop, 2014*; *Pritchard et al., 2010*; *Sella and Barton, 2019*) may explain the allele frequencies of variants associated with complex diseases. In this context, we aim to test the hypothesis that balancing selection is a considerable force in shaping the allele frequencies of extant functional deletions in the human genome.

Balancing selection is a mode of natural selection that maintains a genomic polymorphism by overcoming the stochastic loss or fixation of one of the alleles by genetic drift (*Fijarczyk and Babik, 2015*; *Fisher, 1923*; *Noonan et al., 2006*). HJ Muller was the first to discover balancing selection from his study of balanced lethals in *Drosophila* (*Muller, 1918*). Adaptive variational maintenance by balancing selection may be achieved in a number of ways. In a mechanism known as overdominance (also called heterozygote advantage), the individual who is heterozygous for a certain variant has a higher fitness (*Fisher, 1923*; *Wallace, 1970*). In negative frequency-dependent selection, rarer variants confer higher fitness. This leads to a fluctuation of a variant's frequency in the population until an equilibrium is established, such that neither variant confers an advantage relative to the other (*Smith Maynard et al., 1998*; *Takahashi and Kawata, 2013*). Temporally varying selection, wherein the selection coefficient associated with an allele changes over time, can lead to the oscillation of this allele's frequency over time (*Abdul-Rahman et al., 2021*; *Wittmann et al., 2017*). Spatially varying selection, wherein the selection coefficient associated with an allele varies across geography, may fix this allele locally in one niche and eliminate it in another, leading to the global persistence of variation at the locus (*Hedrick, 2006*; *Levene, 1953*; *Saitou et al., 2021a*).

Unlike positive and negative selection, there is only a modest number of well-established instances of balancing selection (*Charlesworth and Charlesworth, 2017*). In humans, these include polymorphisms of the *ABO* gene, which determine the A, B, and O blood groups (*Ségurel et al., 2012*), and

polymorphisms in the major histocompatibility complex, which encodes cell-surface glycoproteins that display samples of peptides from within the cell on the cell's surface (*Takahata et al., 1992*). Two variants of *ERAP2*, which too is involved in the antigen-presenting pathway, have also been maintained under balancing selection (*Andrés et al., 2010*; *Klunk et al., 2022*). The classic example of recent, shorter-term balancing selection in humans is the maintenance (by overdominance) of sickle-cell alleles at the β-globin locus in the regions of Africa where malaria is endemic (*Allison, 1954a*; *Allison, 1954b*; *Hedrick, 2011*). Similar reasoning applies to certain α-thalassemia alleles in parts of Southeast Asia where malaria is widespread (*Qiu et al., 2013*). In fact, the higher fitness of heterozygotes for thalassemia alleles in malaria-struck regions was presciently predicted by Haldane in 1949 (*Lederberg, 1999*). In the realm of structural variants, complex copy number variation in the human salivary agglutinin genes (*Alharbi et al., 2022*), a regulatory deletion upstream of *APOBEC3* gene family (*Gokcumen et al., 2013*), and a deletion spanning *LCE3B* and *LCE3C* (*Pajic et al., 2016*), which is associated with psoriasis, have been explicitly argued to be evolving in the human lineage under balancing selection.

So far, most systematic investigations into balancing selection in modern humans have focused primarily on genes (*DeGiorgio et al., 2014*; *Soni et al., 2022*) and on single nucleotide variants (SNVs) (*Siewert and Voight, 2020*; *Bitarello et al., 2018*; *Siewert and Voight, 2017*). Additionally, some studies have focused exclusively on 'long-term' balancing selection wherein variants have been maintained in the human lineage since before the split from the chimpanzee clade (*Leffler et al., 2013*). Others have focused on short-term or population-specific balancing selection (*Hedrick, 2011*; *Qiu et al., 2013*). We set out to identify potential targets of balancing selection that are structural in nature and that may not have been captured by earlier studies. Thus, we concentrate our efforts on autosomal deletion polymorphisms (>50 bp) that have been maintained in the human lineage since before the split, approximately ~700,000 years ago, of anatomically modern humans (AMHs) from the lineage that led to both Neanderthals and Denisovans (henceforth, collectively referred to as archaic hominins). In this study, we will use the term 'ancient polymorphisms' to refer to such polymorphisms. Focusing on such 'medium-term' balancing selection will likely allow us to capture more potential targets than could an exclusive study of either 'long-term' or 'short-term' balancing selection. Moreover, deletions are interesting in the context of selection: since a given deletion affects more nucleotides than an SNV, if a defined genomic window is indeed of adaptive importance, deletions may have more profound functional consequences (*Conrad et al., 2010*; *Saitou and Gokcumen, 2020*). Such functional outcomes may translate into non-trivial selection coefficients either for or against the deletion. Additionally, deletions are relatively easier to both genotype and analyze than are other structural variants, making an evolutionary analysis involving deletions tractable.

## Results and discussion

### AMH exhibits a greater proportion of ancient polymorphisms than expected under adaptive neutrality

Older polymorphisms may be more likely than newer ones to exhibit signatures associated with balancing selection because they have survived stochastic fixation or elimination for extended periods. It is, therefore, possible that a certain proportion of human polymorphisms that are older than the AMH-archaic split (~700,000 years), that is, ancient polymorphisms (*Figure 1A*), have been maintained by balancing selection. We tested this hypothesis by comparing the proportion of ancient polymorphisms segregating in AMHs to a neutrally expected distribution of this proportion. If this proportion is significantly higher in the observed data than under the neutrally simulated data, and if we can reject other plausible explanations for difference, we can conclude that some of the polymorphisms older than 700,000 years may have been maintained by balancing selection.

For this test, we focused on 28,291 randomly chosen SNVs (minor allele-count >1) in the Yoruba (YRI) population (*Auton et al., 2015*); see Supplementary material for a discussion of our rationale behind using SNVs rather than deletion polymorphisms, the variant class of our interest. We focused on random SNVs instead of all the SNVs in order to mitigate the biases that would be introduced due to linkage. A variant was classified as ancient if the derived allele was shared, by common descent, with at least one of the four high-coverage archaic hominin genomes (three Neanderthals and one Denisovan) (*Mafessoni et al., 2020*; *Meyer et al., 2012*; *Prüfer et al., 2017*). We found that the

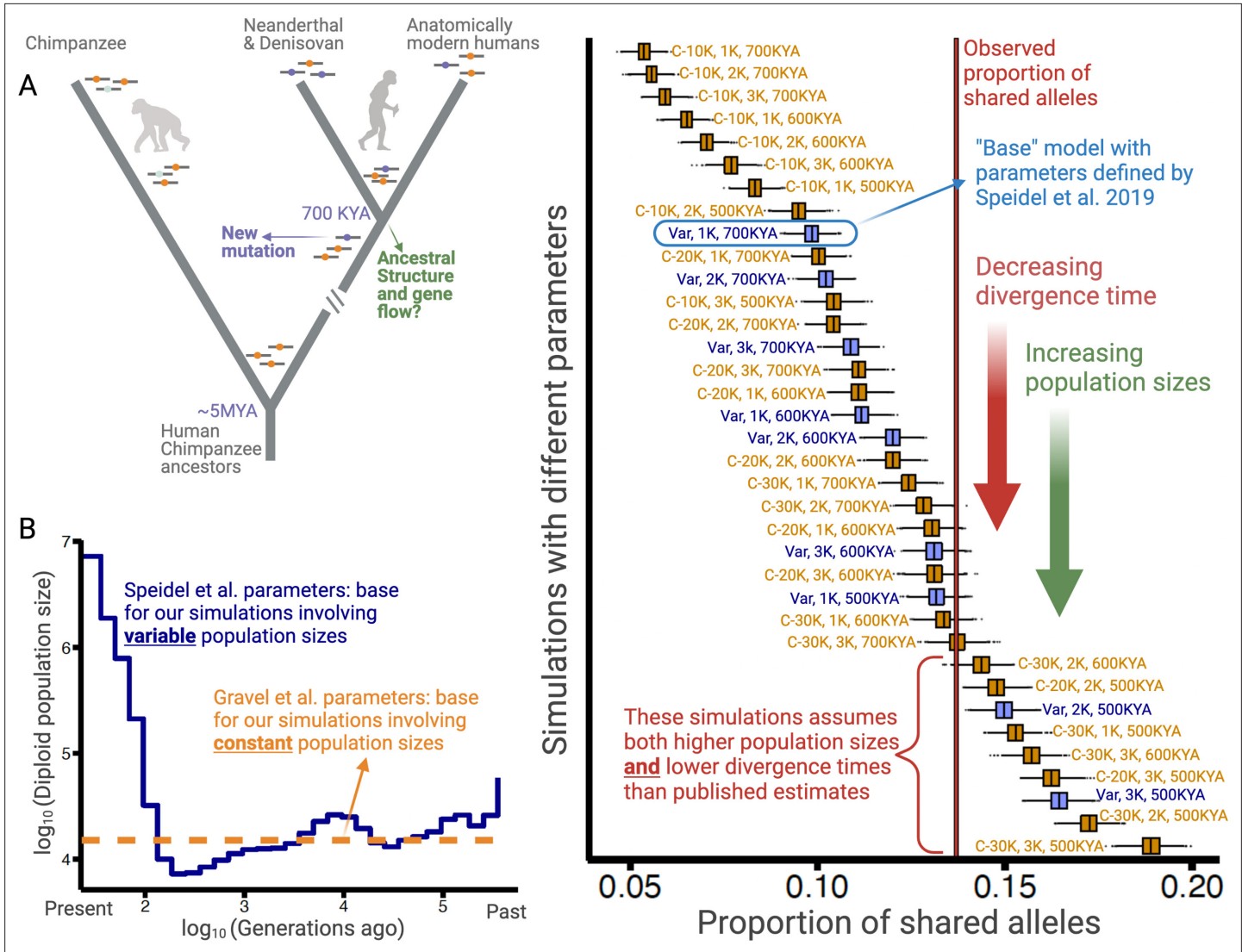

**Figure 1.** Excess of ancient polymorphisms segregating in anatomically modern humans (AMHs). (**A**) A schematic representation of derived 'ancient' variants (purple) that emerged before the AMH-archaic hominin divergence (and after hominin-chimp divergence), and have remained polymorphic in the AMH lineage. The ancestral variants are indicated as orange, and the derived chimpanzee-specific variants are indicated in light blue. (**B**) The Speidel et al. and Gravel et al. simulation parameters. Speidel et al. provide parameters that involve varying population sizes for the YRI population. (**C**) Expected distribution of the proportion of ancient polymorphisms in YRI under different models. Each distribution is labeled with three parameters in the form (*AMH-N_e, Archaic-N_e, time since archaic-AMH divergence*). The simulations where we used variable effective population size published by Spiedel et al. are indicated by blue color and labeled 'Var'. The simulations where *AMH-N_e* is constant are shown in orange, and provide the population size used. The vertical line represents the empirical proportion of ancient polymorphisms in YRI.

The online version of this article includes the following figure supplement(s) for figure 1:

**Figure supplement 1.** Proportion of ancient polymorphisms in observed data (YRI), relative to neutral expectation ('base' model parameters) in various derived allele frequency bins.

**Figure supplement 2.** Simulation results.

ancient SNVs make up 13.7% (3894 SNVs) of the total. Note that we removed the recurrent SNVs from our analysis using a linkage disequilibrium (LD)-based approach (see Methods). To compare the proportion of ancient SNVs against neutral expectations, we used ms (*Hudson, 2002*) to produce 2000 runs of 20,000 neutrally simulated variants in the Yoruba population (see Methods for details). Thereupon, in each run, we calculated the proportion of variants shared with archaic hominins, producing a distribution of the expected proportion of ancient polymorphisms under neutrality.

To ensure that our analysis is not biased by the idiosyncrasies of any particular model, we performed these simulations using 36 distinct models. The models vary by three parameters: $N_e$ of Yoruba/ AMH, $N_e$ of archaic hominins, and the time of divergence between the AMH and the archaic hominin lineage. The $N_e$ for humans can be either constant (ranging from 10,000 to 30,000) or varying over time (*Speidel et al., 2019*; *Figure 1B*). $N_e$ of archaic hominins ranges from 1000 to 3000; and the divergence time ranges from 500 to 700 kya (*Bergström et al., 2021*; *Mafessoni et al., 2020*; *Meyer et al., 2012*; *Prüfer et al., 2014*). In the main text, we focus on the model with variable AMH-$N_e$ (*Speidel et al., 2019*), using the well-accepted archaic hominin $N_e$ of 1000 and a divergence time of 700 kya. We refer to this as the 'base model'.

Using this model, we find that the entire simulated distribution of the proportion of ancient polymorphisms lies to the left of the empirical value of 13.7% (*Figure 1C*). These results hold for all other models with realistic sets of parameters. Even when we change any single parameter to unrealistic levels (e.g., either AMH-$N_e$=30,000, or divergence time = 500 kya), we still observe an excess of ancient polymorphisms. Neutral models can explain the empirical proportion of ancient polymorphisms only when at least two parameters are tweaked in a less realistic direction (e.g., both AMH-$N_e$=30,000 and divergence time = 500 kya). Therefore, we conclude that the high proportion (13.7%) of ancient polymorphisms cannot be explained by realistic neutral scenarios.

There is a possibility that we are observing an empirical excess of ancient polymorphisms owing to a high incidence of recurrent mutations in the AMH and archaic hominin lineages that remained undetected by the LD-based approach we used to identify them. Since CpG sites may be particularly prone to recurrent mutations, we calculated the proportion of empirical ancient polymorphisms again using only A$\rightleftarrows$T SNVs. This analysis yielded a proportion of 14.24%, not much different from the previously calculated 13.7%. Moreover, if recurrence was the main cause of the observed excess of ancient polymorphisms, we would expect this excess (relative to the simulated distribution) to be more pronounced among polymorphisms with low derived allele frequencies. To test if this is the case, we repeated our analysis using the base model for simulations, dividing the empirical and simulated SNVs into derived allele frequency bins (*Figure 1—figure supplement 1*). We observed that the excess of ancient polymorphisms is, in fact, most pronounced at high derived allele frequency. Both these results combined suggest that our results are not biased due to undetected recurrent mutations.

Next, we consider possible non-adaptive explanations for the observed excess of ancient polymorphisms. First, we considered scenarios invoking structure in the population that was ancestral to both AMHs and archaic hominins, while allowing gene flow between the latent subgroups within the ancestral population (*Figure 1*; see Methods for details). We found that the excess of ancient polymorphisms can be explained by structuring the ancestral population into three distinct subpopulations, such that the fraction of each subgroup formed by the migrants from each of the other subgroups, every generation, is below 0.0075% (*Figure 1—figure supplement 2A–B*). However, the allele frequency spectrum for SNVs simulated with ancient population structure significantly deviates from the observed allele frequency spectrum in that the former overestimates the intermediate/common variants (*Figure 1—figure supplement 2C*). Therefore, invoking such structure to explain the excess of ancient polymorphisms may be unrealistic. Another possible explanation comes from the evidence of introgression from early modern human ancestors into Neanderthals to the exclusion of Denisovans (*Posth et al., 2017*). Such admixture can increase the apparent proportion of ancient polymorphisms due to elevated allele sharing with Neanderthals. However, we do not observe a higher proportion of derived allele sharing (by common descent) with Neanderthals than with Denisovans. In fact, the proportion of derived alleles shared with the Denisovan (9.75%) and the Altai Neanderthal (10.28%) is higher than the proportion shared with Chagyrskaya and Vindija Neanderthals (6.14% each), which is incompatible with such introgression as the prime cause of excessive ancient polymorphisms in AMHs. We note that the differential allele sharing with the Denisovan and Altai Neanderthal on one hand, and the Vindija and Chagyrskaya Neanderthal on the other would be an interesting subject for future studies.

Overall, based on our current knowledge of ancient interactions and demographic history, our analyses implicate balancing selection as a possible cause of the excess of observed ancient polymorphisms. Next, we focus on deletion polymorphisms segregating in AMHs, categorize them based on their evolutionary histories, and test whether ancient deletion polymorphisms are enriched for targets of balancing selection.

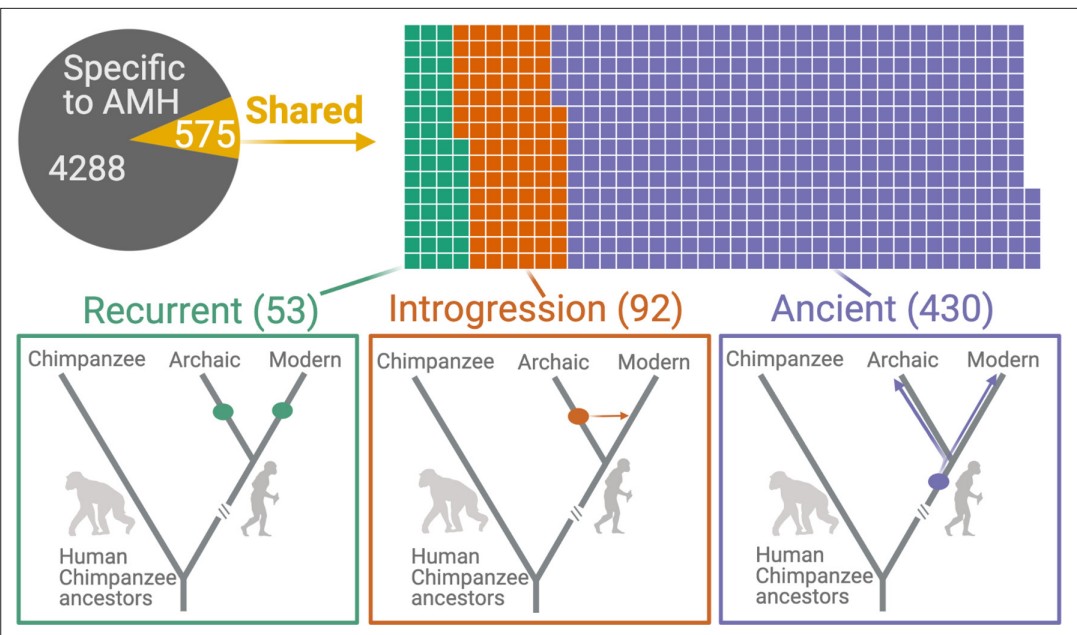

**Figure 2.** Deletions in anatomically modern humans (AMHs) that are shared with archaic hominins. The top panel shows the categorization of deletion polymorphisms as AMH-specific, recurrent (green), introgressed (orange), or ancient (purple). The evolutionary histories of shared deletions are summarized schematically in the bottom panel.

The online version of this article includes the following figure supplement(s) for figure 2:

**Figure supplement 1.** Read depth-based pipeline to identify deletions in archaic hominin genomes: Distribution of the modified Z-score of the read depth across the 32,154 biallelic anatomically modern human (AMH) deletions in the archaic genomes.

## Categorizing human deletions based on their evolutionary history

Having established that AMHs exhibit an excess of ancient polymorphisms that cannot be explained solely by non-adaptive causes, we identify ancient deletion polymorphisms among AMHs. Since the vast majority of deletions in AMHs are derived relative to chimpanzees (Supplementary material), this could be accomplished by identifying AMH deletions that are shared with archaic hominins by common descent.

In this analysis, AMHs are represented by the YRI (Yoruba), CEU (Utah residents with Northern and Western European ancestry), and CHB (Han Chinese in Beijing) from 1000 Genomes Project Phase 3 dataset (*Auton et al., 2015*); and archaic hominins are represented by the four available high-coverage (~30×) archaic hominin genomes (*Mafessoni et al., 2020*; *Meyer et al., 2012*; *Prüfer et al., 2017*). Our choice of AMH populations was guided by our wish to both sample from different regions, and use relatively well-studied populations. We genotyped all AMH deletions in the archaic hominin genomes using a read depth-based pipeline (*Supplementary file 1*). We considered a deletion 'shared' if it was identified in at least one of the four archaic genomes. For our analysis, we used only the deletions with an allele-count >1 in YRI, CEU, and CHB combined. Additionally, we retained only 4863 human deletion polymorphisms that are in LD ($r^2$ >0.9) with at least one SNV (*Supplementary file 2*). We imposed this LD requirement because SNVs in LD with the deletion can enable us to distinguish the shared deletions that are introgressed or recurrent from those that are shared by common descent.

We found that 575 (11.8%) AMH deletions were shared with archaic hominins, that is, identified in at least one archaic hominin genome (*Figure 2*). We identified 53 instances of independent emergence (*recurrent deletions*) in archaic hominin and AMH lineages, wherein no SNV that is in LD with the deletion in AMHs accompanied the deletion in archaic hominins. In parallel, we identified 92 deletions that were *introgressed* from archaic hominins into AMHs: the SNVs in LD with these deletions were present in previously identified introgressed haplotypes (*Taskent et al., 2017*; *Vernot and Akey, 2014*). By this process of elimination, we found that 430 (8.8% of the total) shared deletions are

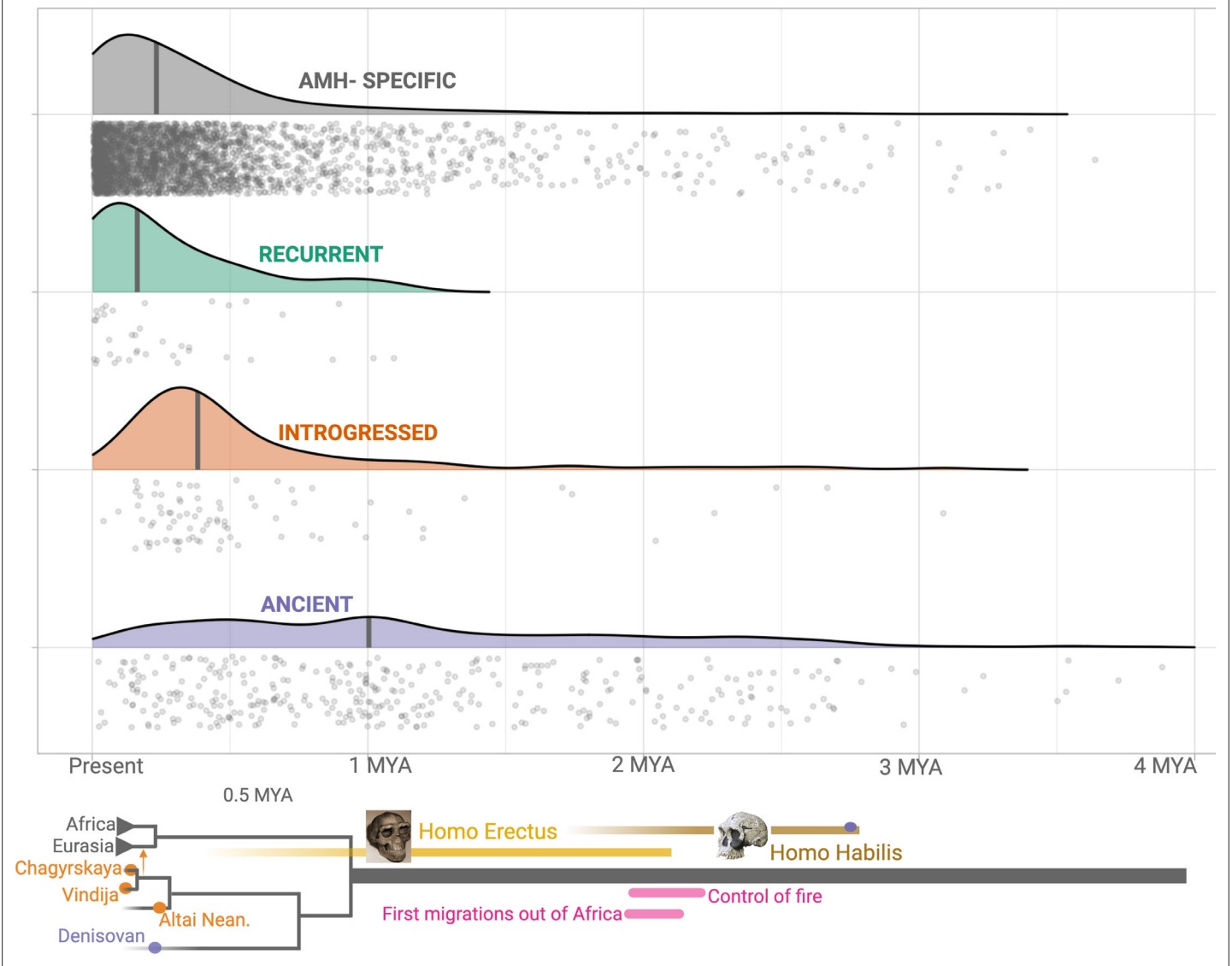

**Figure 3.** Age estimates of the haplotypes harboring polymorphic deletions. The x-axis shows the age estimates, obtained using Relate, for the deletions. For orienting the reader regarding the age of these variants, we provide below a schematic phylogeny representing recent human evolution.

The online version of this article includes the following figure supplement(s) for figure 3:

**Figure supplement 1.** GEVA ages of deletions across categories.

ancient polymorphisms, that is, they are shared with archaic hominins by common descent and thus emerged at least ~700,000 years ago.

To confirm that our pipeline for identifying ancient deletions (*Figure 2—figure supplement 1*) has high accuracy, we estimated the ages of deletions, based on the ages of SNVs in LD. We used two methods in parallel: (1) Human Genome Dating (*Albers and McVean, 2020*) and (2) Relate (*Speidel et al., 2019*) (see Methods). If both our genotyping pipeline and categorization of shared deletions (as recurrent, introgressed, or ancient) are sound, we should expect that ***Age***(*human-specific*) ≈ ***Age***(*recurrent*)<***Age***(*introgressed*)<***Age***(*Ancient*). Both methods yielded the expected pattern of ages across the categories of deletions (*Figure 3*, *Figure 3—figure supplement 1*). We found that the median age for ancient deletions, using Relate, is ~1 million years. About 15% (63) of these deletions are older than 2 million years. In contrast, the median age of non-ancient deletions is ~239,000 years. As such, we infer that both our genotyping pipeline and deletion categorization approach are sound. Counterintuitively, a small number of 'ancient' deletions have very recent dates. This may be due

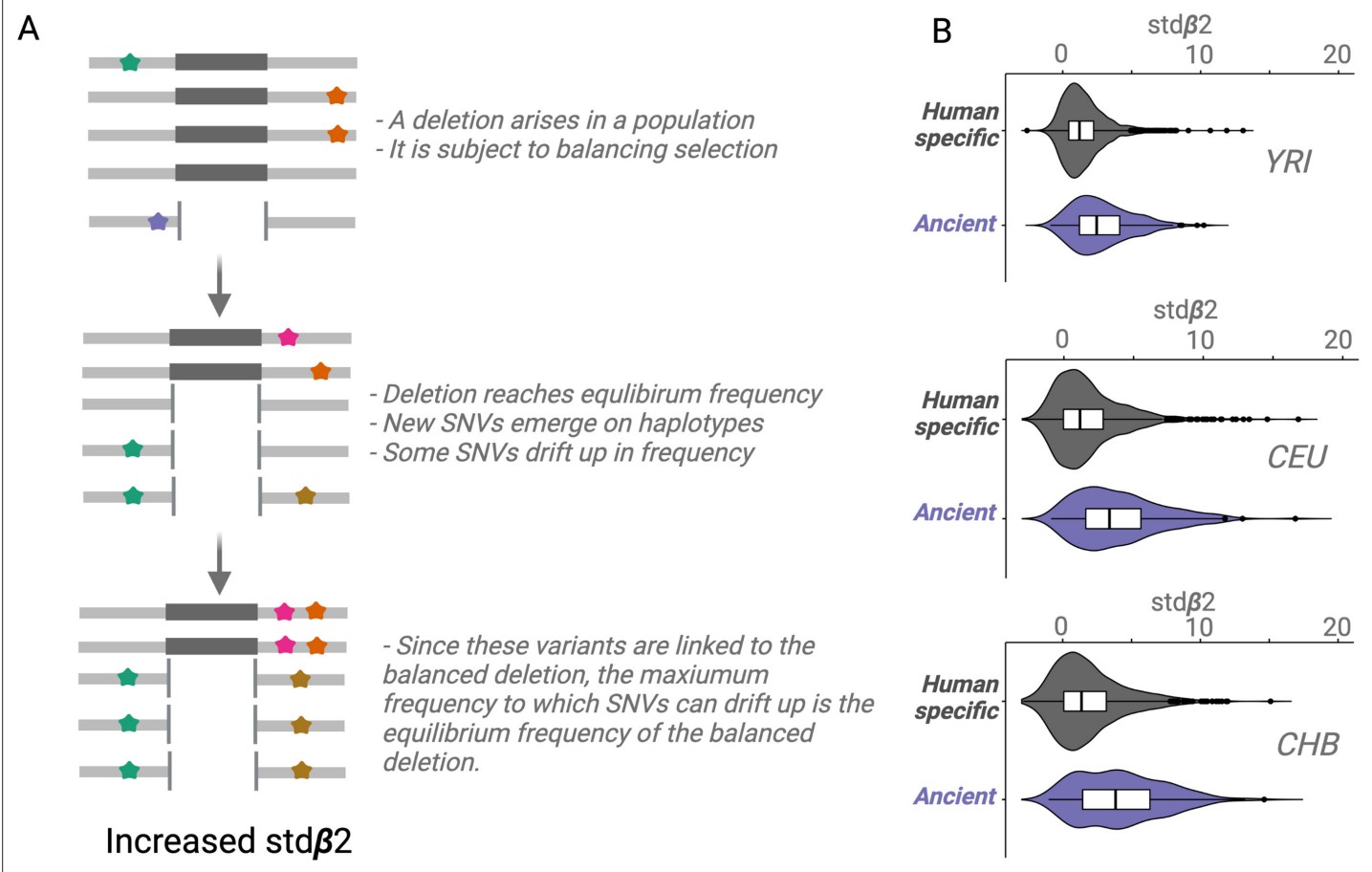

**Figure 4.** An empirical assessment of putative balancing selection among ancient deletions. (**A**) The conceptual framework in which stdβ2 statistic works. The last step demonstrates 'Goldilocks' drift (the process that results in allelic class build-up). (**B**) A box plot for stdβ2 for anatomically modern human (AMH)-specific, versus ancient deletions (frequency >5% in respective populations). Higher stdβ2 values for older deletions represented in purple empirically show that older deletions are significantly enriched for targets of balancing selection. All comparisons are significant, $p < 10^{-7}$ (Wilcoxon).

to instances of recent soft sweeps involving some deletions leading to an increased length of the associated haplotype and an artificial decrease in age. Second, some ancient deletions may have low frequencies, which too, creates a downward bias in age. Lastly, this may be due to miscategorization of non-ancient deletions as ancient. Next, we test if ancient deletions are enriched for targets of balancing selection relative to non-ancient deletions.

## Ancient deletion polymorphisms are more likely to be targets of balancing selection than are non-ancient ones

We used the stdβ2 statistic (**Siewert and Voight, 2020**) to test the hypothesis that ancient deletion polymorphisms are more likely than are non-ancient deletions to be targets of balancing selection. Stdβ2 is a measure of balancing selection, that calculates the weighted average of the number of flanking-derived variants, where weights are the similarity in frequency between the core allele and the flanking variants (**Siewert and Voight, 2017**).

For a conceptual understanding of Stdβ2 (**Figure 4A**), suppose a deletion emerges and the resulting polymorphism is subject to balancing selection; the deletion will rise in frequency until it reaches a certain equilibrium frequency. New SNVs will emerge on the haplotypes carrying the deletion. Some of these SNVs will drift upward in frequency, but since these SNVs will be linked to the deletion, they too can only rise to the equilibrium frequency of the balanced deletion (**Siewert and Voight, 2020**; **Siewert and Voight, 2017**). We refer to this type of drift as *Goldilocks drift*, since the linked SNVs drift upward to the 'just-right' equilibrium frequency of the balanced deletion. Goldilocks drift thus leads to allelic class build-up (analogous to how hitchhiking leads to sweeps), which refers

to a situation involving the fixation of many flanking variants within the set of haplotypes carrying the deletion. Stdβ2 value for a core variant may be thought of as the average intensity of *Goldilocks drift* experienced by SNVs around it. Therefore, a high stdβ2 value for a variant implies that it is either a target of balancing selection or close to a target of balancing selection.

We observed that stdβ2 estimates for ancient deletions are significantly larger than those for non-ancient deletions across YRI, CEU, and CHB populations ($p < 10^{-7}$, Wilcoxon) (*Figure 4B*). These results provide empirical evidence that ancient deletion polymorphisms are enriched for targets of balancing selection. Our results are consistent with other recent studies (*Soni et al., 2022*) that have argued that the role of balancing selection in explaining the maintenance of common variation in the human lineage is underappreciated.

Previous genome-wide balancing selection scans focused on either individual genes or SNVs. Consequently, we do not expect to find many ancient deletions that have previously been reported as targets of balancing selection. Nevertheless, we investigated whether the exons in any of the genes that have been reported as targets of balancing selection in *DeGiorgio et al., 2014*, or *Soni et al., 2022*, overlap with ancient deletions. We found no overlaps. We also found that 77 common (>5% in YRI, CEU, and CHB combined) ancient deletions were in LD ($r^2 > 0.9$) with SNVs that had high (in the 95th percentile) Stdβ2 (*Siewert and Voight, 2020*) values in YRI, CEU, and CHB. This is unsurprising since this is the measure we used to show that ancient deletion polymorphisms are enriched for balancing selection targets. Interestingly, one of the ancient deletions with a high associated Stdβ2 value overlaps a candidate region for balancing selection previously identified using the non-central deviation method (*Bitarello et al., 2018*). This 433 bp deletion (esv3607090), which is 2 million years old and common across populations, deletes part of an intron of the *STK32A* gene. This could be an interesting subject for future studies. Regardless, a vast majority of common ancient deletions (73%) were not reported previously as balancing selection candidates and thus are novel targets for future studies.

## Phenotypic relevance of ancient deletion polymorphisms

Selection can only act on a region of the genome by means of the phenotypic function it confers. It follows then that any adaptively maintained ancient polymorphisms must be functional. If an appreciable proportion of ancient deletion polymorphisms have evolved under balancing selection and more recent deletion polymorphisms have not, we should expect ancient deletions to be enriched for functional effects. To avoid biases introduced by different proportions of rare variants among ancient versus non-ancient deletions, we focus only on deletion with frequency >5% in AMHs. For both ancient and non-ancient deletions, we investigated (1) whether a deletion intersects exons and (2) whether any of the SNVs in LD with it are associated with UK BioBank genome-wide association

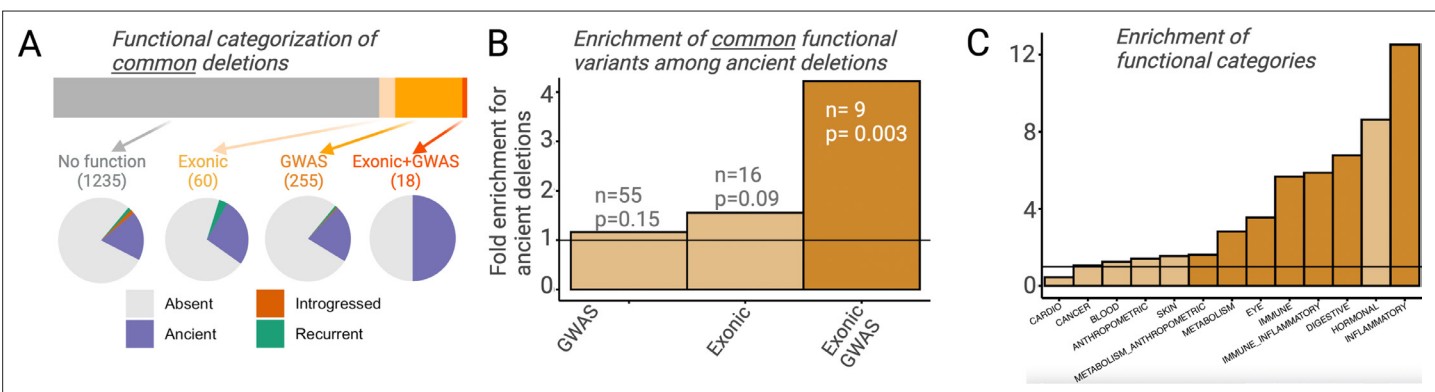

**Figure 5.** Functional enrichment among ancient deletions. (**A**) Functional categorization of common deletions. Within each category, the proportions of deletions falling under different evolutionary categories are shown in pie charts. (**B**) Permutation-based analysis of enrichment of functionality among ancient deletions, relative to non-ancient deletions. The black horizontal line indicates the expected ratio of 1.0. For each definition of functionality, the number of functional ancient deletions, and the p-value associated with the enrichment are provided. (**C**) Permutation-based enrichment analysis for different phenotypic categories (based on genome-wide association studies [GWAS]) among ancient deletions, relative to non-ancient deletions. The black horizontal line indicates the expected ratio of 1.0. Dark orange indicates a statistically significant deviation from the expected ratio of 1.0. Light orange means no significant deviation from the expected ratio of 1.0.

studies (GWAS) traits with p<10⁻⁸ (http://www.nealelab.is/uk-biobank/; *Figure 5A*; see Methods). We did not observe a significant increase in either the proportion of exonic (ancient = 5.6%; non-ancient=3.6%) or GWAS-associated (ancient = 19.1%; non-ancient=16.4%) ancient deletion, relative to non-ancient deletions (*Figure 5B*). However, when we classify a deletion as functional more conservatively, that is, it both intersects an exon and has a GWAS association (i.e., one of the SNVs in LD with it has a GWAS association), we observe a 4.2-fold enrichment (p=0.003) of functional variants among ancient deletions (*Figure 5B*). In fact, out of the 18 common deletions (frequency >5%) that both intersect genes and have GWAS associations, 9 (50%) are ancient. Further, we observed a phenotypic enrichment among ancient deletions for some GWAS trait categories: a 12.5-fold enrichment (p=10⁻⁵) of traits related to inflammatory response and a 2.8-fold enrichment (p=0.003) of traits related to metabolism (*Figure 5C*).

A focused literature review and analysis of functional effects associated with some of the ancient deletions revealed multiple mechanisms through which they affect function (*Figure 6A*). First, whole-gene deletions may affect the function of entire environment-interacting gene families. We found two ancient whole-gene deletions: esv3587563 (deleting *LCE3B* and *LCE3C*) and esv3600896 (deleting *UGT2B28*). The members of the *LCE3* and *UGT2B* gene families mediate immune response and steroid metabolism, respectively; genes from both families likely evolved under adaptive forces (*de Guzman Strong et al., 2010*; *Pajic et al., 2016*; *Starr et al., 2021*; *Xue et al., 2008*). The functional consequence of whole-gene deletions is, of course, loss of function of the deleted genes. In addition, esv3587563 is associated with an increase in the expression of *LCE3A* (*de Guzman Strong et al., 2010*; *Pajic et al., 2016*), while esv3600896 is associated with an increase in the expression of *UGT2B11*. Thus, we propose that whole-gene deletions of members of environment-interacting gene families may lead to the functional 'fine-tuning' of the entire gene family.

Second, we revealed dozens of potentially adaptive ancient deletions that both mediate gene regulation and are associated with human traits. For example, we found multiple ancient deletions that are proximal to the *HLA* locus, associated with immune-related phenotypes, and affect the expression levels of nearby genes. One such example, the deletion esv3608584, is noteworthy within the context of balancing selection because it affects the expression of different *HLA* genes in opposite directions (*Figure 6B*). As such, this deletion may lead to increased susceptibility to some pathogens while increasing the defenses against others. Further, we observed that the ancient deletions in the *HLA* locus also lead to the expression of different isoforms of HLA genes. Using the GTeX database, we found at least four other instances where ancient deletions lead to the expression of different isoforms, including deletions affecting the *HLA-DRB1-6, HLA-DOB, SIRPB1, GHR,* and *CYP3A43* genes. We recently showed that the ancient deletion of the third exon of the growth hormone receptor gene leads to the expression of a smaller version of growth hormone, which may be adaptive in times of starvation (*Saitou et al., 2021b*). The *SIRPB1* gene encodes a glycosylated transmembrane receptor protein (*Kharitonenkov et al., 1997*), and its different isoforms may lead to the recognition of different pathogens. Similarly, *CYP3A43*, a member of the cytochrome p450 gene family, is involved in metabolizing external substances, and genetically determined isoforms contribute to its functional variation in humans (*Agarwal et al., 2008*). Thus, ancient deletions that lead to specific isoform expression may have been adaptively evolving to adjust the function of environment-interacting genes across both geography and time. It is important to acknowledge that these non-exonic deletions may not be the causal variant in the associated haplotypes. Nevertheless, the full extent of deletion polymorphisms shaping the expression levels and sculpting the isoform diversity at the genetic level remains a fascinating area of future research.

## The effect of negative selection is stronger on deletions than SNVs

Based on previous work, we expect that deletions are more likely than SNVs to be under negative selection (*Conrad et al., 2006*; *Kondrashov, 2017*; *Lin et al., 2015*; *Lin and Gokcumen, 2019*). To investigate the magnitude of this effect, we compared the proportion of SNVs and deletion polymorphisms that are ancient. Applying the same bioinformatic pipeline to identify ancient polymorphisms in both cases, we found that 13.7% of SNVs and 9.6% of deletion polymorphisms are ancient in YRI. This result alone suggests that deletion polymorphisms are more likely than SNVs to be eliminated by negative selection, a trend that we expect to be more pronounced with increasing ages of polymorphisms. The greater intensity of negative selection acting on deletions implies that deletions

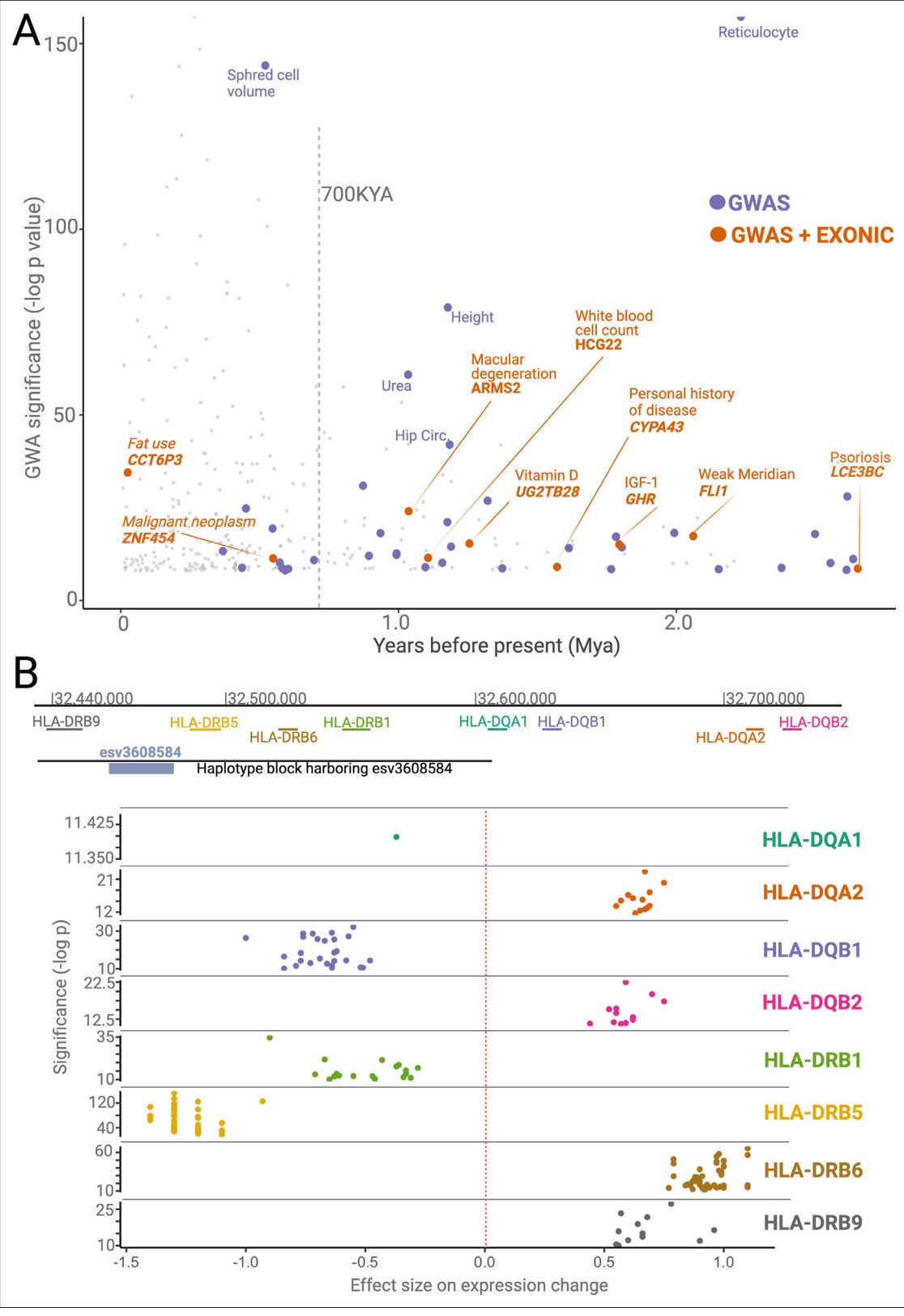

**Figure 6.** Phenotypic effects associated with deletions. (**A**) The significance levels (-log(p-value)) of phenotypic associations of deletions with genome-wide association studies (GWAS) traits as a function of their emergence time. Gray points indicate non-ancient deletions. Purple and orange points indicate non-exonic ancient deletions with GWAS hits and exonic ancient deletions with GWAS hits, respectively. The genes whose exons are covered by ancient deletions, and the traits associated with ancient deletions are mentioned in the plot. (**B**) The significance levels (-log(p-value)) and sizes of

*Figure 6 continued on next page*

*Figure 6 continued*
expression level changes of nearby *HLA* genes associated with the presence of the deletion esv3608584. Each color refers to a different *HLA* gene. Each point in a given color represents a different tissue. Only those tissues whose expression level changes are statistically significant are shown here.

are, in general, more deleterious than SNVs. It follows that younger (non-ancient) deletions currently segregating in human populations, which negative selection has not yet purged, are more likely to be deleterious (and perhaps disease-causing) than SNVs are (*Kondrashov, 2017*).

The preceding argument makes intuitive sense since a given deletion spans more bases than does an SNV. If this intuition is correct, we expect that larger deletions should, on average, experience more intense negative selection. Since most large ancient deletions would have been purged by negative selection, we expect surviving ancient deletions to be, on average, smaller than non-ancient deletions. We test this using common deletions (frequency >5% in YRI, CEU, and CHB combined). Ancient deletions are indeed, on average (median), 14% shorter than non-ancient deletions (p=0.02; permutation test). Nevertheless, there is an excess of long deletions among ancient deletions relative to non-ancient deletions (*Figure 7A*). In particular, the 95th and 98th size percentiles of ancient deletions are 33% (p=0.04; permutation test) and 128% (p=0.005; permutation test) larger than non-ancient deletions, respectively (see Methods). This excess of longer deletions is inconsistent with evolution under neutrality or negative selection. Therefore, the longest 5% of ancient deletions are excellent targets for future studies of balancing selection. In fact, three out of the four GWAS-associated common exonic deletions intersecting *SIRPB1*, *LCE3A*, *LCE3B*, and *UGT2B28* are in the 95th percentile of the size distribution of ancient deletions.

## Strong overdominance is rare among ancient deletion polymorphisms

Having established that ancient deletion polymorphisms appear enriched for targets of balancing selection, we wanted to investigate whether classical overdominance is a common mechanism underlying this observation. To accomplish this, we first identified the genomic signatures that we expect to see in a region where a polymorphism has evolved under overdominance, and then we looked for these signatures among ancient deletions. To identify the signatures of overdominance, we simulated

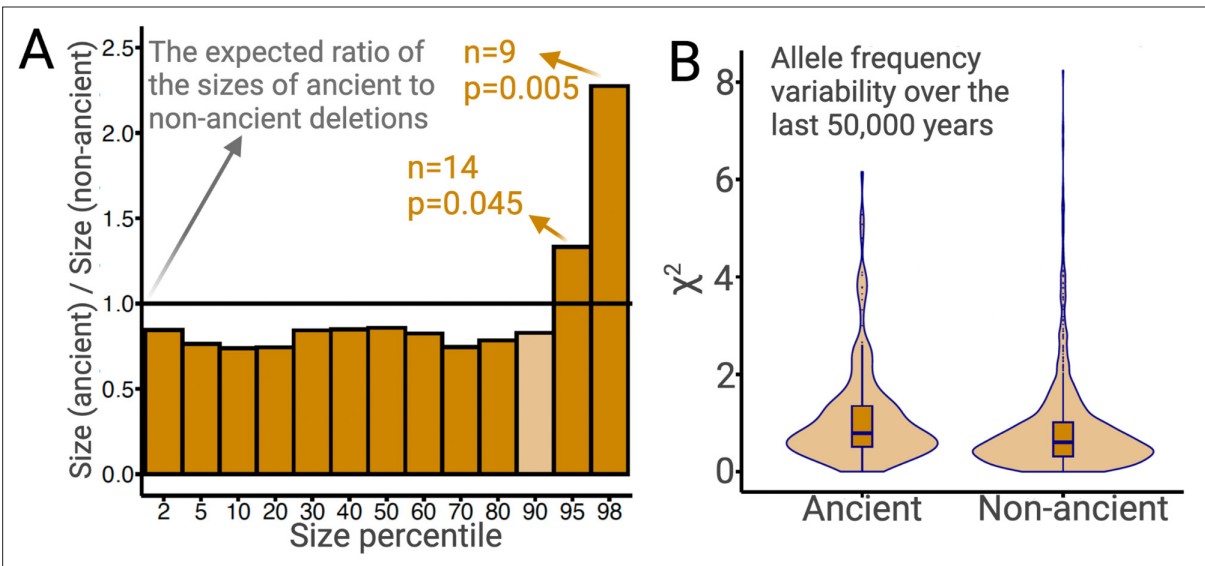

**Figure 7.** Ancient versus non-ancient deletions. (**A**) The ratios of sizes of ancient deletions to those of non-ancient deletions at different size percentiles. The black horizontal line refers to the expected ratio of 1.0. Dark orange bars refer to a statistically significant (permutation test) deviation from the expected ratio. Light orange bars mean that the deviation from the extend ratio of 1.0 is not statistically significant. (**B**) The estimated measure of allele frequency change ($\chi^2$) between 50,000 and 5000 years before present in common ancient versus common non-ancient deletions. Ancient deletions have significantly (p=2 × 10⁻⁷, Wilcoxon) higher frequency variability over the last 50,000 years.

The online version of this article includes the following figure supplement(s) for figure 7:

**Figure supplement 1.** Effects of negative selection and overdominance.

sequence evolution under neutrality and overdominance (using a variety of selection coefficients), in turn, for variants that emerged one million years ago. We asked whether we can distinguish between neutrality and overdominance by calculating several population genetic statistics, including Tajima's D, π, θ, etc., on sequences generated from the neutral and overdominance simulations (see Methods for full list). We found that none of these statistics alone can distinguish overdominance from neutrality, even for strong selection coefficients (*Figure 7—figure supplement 1*).

Instead, we found that a distinct feature of overdominance is that the allele frequency rapidly increases (similar to a selective sweep) until it reaches an equilibrium frequency, whereafter it remains remarkably stable across time (*Figure 7—figure supplement 1*). In contrast, under neutrality, a random change in allele frequency in every generation produced elevated noise across time in allele frequency trajectories. To ascertain if overdominance is a common mechanism of evolution for ancient deletions, we inferred the allele frequency trajectories of ancient deletions using *Relate* (*Speidel et al., 2019*) and quantified the variation in allele frequency between 5000 and 50,000 years ago by squaring the standardized allele frequency difference ($\chi^2$) (Methods). We already know that ancient deletions are enriched for targets of balancing selection. If a large proportion of these balancing selection targets have evolved under overdominance, we expect ancient deletions to have more stable allele frequencies across time, relative to non-ancient deletions, leading to smaller $\chi^2$ values on average. However, we do not observe this trend among common (frequency >5% in YRI, CEU, and CHB combined) deletions (*Figure 7B*). Consequently, at least with our current resolution of allele frequency trajectory estimation, we found no evidence for overdominance being the prime mode of balancing selection operating on ancient polymorphisms in AMHs.

In fact, we observe that ancient deletion polymorphisms exhibit greater allele frequency variation than do non-ancient deletions (*Figure 7B*; p=$2 \times 10^{-7}$, Wilcoxon). This suggests that a large proportion of the instances of medium-term balancing selection likely involve temporally and spatially variable selection, which lead to elevated levels of allele frequency variation over time. This is consistent with our locus-specific analyses of ancient deletion polymorphisms. For example, we recently reported that the deletion (esv3604875) of the third exon of the human growth hormone receptor gene (*GHR*) has evolved under temporally and geographically variable adaptive constraints (*Saitou et al., 2021b*). In fact, this deletion is in the 93rd percentile of $\chi^2$ values. Ancient deletions like these are common and old, and also exhibit high population differentiation. Collectively, we argue that such adaptive maintenance of ancient, functional alleles may be due to selection constraints that vary across time and geography.

## Conclusion

This manuscript asked whether adaptive forces have maintained ancient deletion polymorphisms in humans. We provide evidence supporting the idea that AMHs exhibit an excess of ancient polymorphisms, relative to the neutral expectation. Using simulations and empirical data, we provide evidence for the notion that balancing selection is likely a considerable force in the evolution of deletion polymorphisms. We show that when functionality is defined conservatively, ancient deletions are disproportionately functional, compared to non-ancient deletions. In fact, 50% of such functionally relevant deletions are ancient. Additionally, ancient deletions are enriched for associations related to metabolism and inflammatory response. Our results suggest that classical overdominance may not be the prime mode of balancing selection affecting the evolution of ancient deletions. Instead, geographically and temporally variable, as well as frequency-dependent selection may underlie the maintenance of ancient functional deletions. We also provide insights about the mechanisms by which a deletion could confer function: in addition to previously defined functional effects of deletions such as the loss of a gene's function and regulation of expression levels, we highlight multiple instances where the presence of ancient deletions leads to the expression of different isoforms. Overall, our study contributes to the growing body of evidence supporting the notion that balancing selection may be an important force in the evolution of genomic variation shared among human populations. These ancient variants are an important part of our legacy as a species: something we all share.

## Methods

### Proportion of ancient polymorphisms in neutral simulations versus observation

We compared the proportion of ancient polymorphisms in randomly chosen YRI SNVs against the expected distribution of this proportion under neutrality. We obtained SNVs from the 1000G phase-3 vcf files (*Auton et al., 2015*) for the analysis described above. Using a script written in AWK (*Aho et al., 1979*), we subsetted the vcf files to retain only those biallelic SNVs that contained information about the ancestral/derived status of the two alleles. We used the `--keep` option in vcftools (*Danecek et al., 2011*) to retain individuals only from the Yoruba population. We then used the `--mac` option to retain only those SNVs for which the minor allele-count was greater than 1. The allele-count filter was used to exclude singletons which could create spurious results for the LD analysis described below. On the resulting vcf files, we used the SelectVariants tool with the `--select-random-fraction` option in GATK (*Van der Auwera and O'Connor, 2020*) to randomly retain 0.25% of the variants. This resulted in a set of 38,231 SNVs. Next, we investigated whether the random SNVs are in LD with any other SNVs in their vicinity. In particular, we used the `--hap-r2-positions` and `--min-r2` option in vcftools, along with the ancestral/derived status of alleles, to retain only those random SNVs wherein the derived allele was in LD ($r^2 > 0.9$) with the derived allele of another SNV within a 50 kb radius of the random SNV. This resulted in 28,491 SNVs. We only retained the random SNVs with variants in LD in their vicinity to rule out cases of recurrence of the SNV between AMHs and archaic hominins, as described below. There seems to be one bias that may be introduced by eliminating polymorphisms with no SNVs in LD in AMHs: this would bias our analysis to regions with low recombination rates. In these regions, we would expect higher background selection due to the Hill-Robertson effect, leading to a deflation in the proportion of ancient polymorphisms. Since we observe a larger than expected proportion of ancient polymorphisms despite this bias, we can conclude that this bias only makes our analysis more conservative.

We then inspected the 28,491 random SNVs to see whether the derived alleles are shared with any of the four high-coverage archaic genomes (Altai Neanderthal, Vindija Neanderthal, Chagyrskaya Neanderthal, and the Denisovan). We found that 4616 SNVs (16.2%) had their derived allele shared (either homo- or heterozygously) with at least one of the four archaic hominins. Since we are only interested in polymorphisms older than 700,000 years, we want to focus only on SNVs where the derived allele is shared with archaic hominins by *common descent*. We thus excluded the SNVs where the derived allele emerged independently (recurrence) in AMHs and archaic hominins in the following way. For each of the 4616 core SNVs with shared derived alleles in archaic hominins, we tested whether any of the derived alleles in LD with the core derived allele is also present in any of the same archaic hominins which carry the derived allele for the core SNV itself. If any of the archaic hominins contain both the derived allele of the core SNV along with at least one derived allele in LD with the derived allele of the core SNV, we classify that core SNV as 'shared by common descent'. If any core SNV has a derived allele shared with archaic hominins but not a derived allele in LD with the core SNV, we classify the core SNV as 'recurrent'. This approach yielded 3894 SNVs (13.7%) wherein the derived allele is shared with at least one of the archaic hominins by common descent.

In order to investigate whether this percentage (13.7%) of ancient polymorphisms (700,000 years old polymorphisms) is significantly higher than the neutral expectation, we wanted to calculate the same percentage for a set of neutrally simulated SNVs. We used the program ms (*Hudson, 2002*) to neutrally simulate a set of 20,000 (independent, and therefore freely recombining) variants 2000 times using various models. All of these models included 216 haploid genomes representing YRI (matching the YRI sample size in 1000G dataset) and 2 haploid genomes representing each of the four archaic hominins. For every model, and for each of the 2000 runs we calculated the proportion of SNVs present in the YRI with minor allele-count >1 that were shared with at least one of the archaic hominins. Thereby, we obtained a distribution of the proportion of ancient polymorphisms under each of the models.

We used a total of 36 models. The models varied by three parameters: $N_e$ of Yoruba/AMH, $N_e$ of archaic hominins, and the time of divergence between the AMH and archaic hominin lineages. For 27 of these models, $N_e$ for humans was constant across time (ranging from 10,000 to 30,000 across models); for 9, it varied over time (*Speidel et al., 2019*; *Figure 1B*). The $N_e$ of the archaic hominin lineage was constant over time in each model ranged from 1000 to 3000 across models; and the time

of divergence between AMH and archaic hominins ranges from 500 to 700 kya across models (**Bergström et al., 2021**; **Mafessoni et al., 2020**; **Meyer et al., 2012**; **Prüfer et al., 2014**). In the main text, we focused on the model with variable AMH-$N_e$ (**Speidel et al., 2019**), using the well-accepted archaic hominin $N_e$ of 1000 and a divergence time of 700 kya; we referred to this as the 'base model'. For each model, we assumed that Denisovans diverged from Neanderthals ~400,000 years ago, the Altai Neanderthal lineage separated from the Vindija-Chagyrskaya lineage ~130,000 years ago, and the Vindija lineage separated from the Chagyrskaya lineage ~90,000 years ago (**Bergström et al., 2021**; **Gravel et al., 2011**; **Mafessoni et al., 2020**; **Meyer et al., 2012**; **Prüfer et al., 2014**). Moreover, the generation time was assumed to be 29 years (**Fenner, 2005**; **Langergraber et al., 2012**; **Li and Durbin, 2011**).

To test if the empirical excess of ancient polymorphisms is more pronounced among low derived allele frequency variants (due to recurrence), we repeated our analysis using the base model (variable AMH $N_e$, Archaic $N_e$ = 1000, divergence time = 700 kya) for simulations, dividing the empirical and simulated SNVs into derived allele frequency bins. To ensure there are enough variants in each frequency bin, we used a larger set of randomly chosen YRI SNVs. In particular, we used 300,000 SNVs that have a minor allele-count >1 and at least one variant in LD ($r^2$ >0.9) with them. Using the same method as described above, we identified the SNVs that are shared with archaic hominins by common descent. We used the base model to simulate 1500 runs of 400,000 SNVs. Thereupon, we divided the empirical and simulated SNVs into 10 derived allele frequency bins of uniform length, and compared the simulated distribution of the proportion of ancient polymorphisms with the observed proportion in each bin (**Figure 1—figure supplement 1**). Moreover, to gauge whether recurrence at CpG sites leads to the excess of ancient polymorphisms, we subsetted the 300,000 SNVs to retain only the 21,402 A⇌T SNVs, and calculated the proportion of ancient polymorphisms therein.

Additionally, we also performed another set of neutral simulations, this time with structure introduced in the population ancestral to the archaic hominins and AMHs. This too was done using Hudson's ms. This was done using a constant size model with YRI/AMH $N_e$ = 14,474 (**Gravel et al., 2011**), Archaic $N_e$ = 1,000, and divergence time = 700 kya. The effective population size for each of the subgroups in the population ancestral to both AMHs and archaic hominins was set to 10,000. We define $m_{i,j}$ as the fraction of subgroup *i* that is formed by the migrants of subgroup j in each generation, where i ≠ j and i,j ∈ {1,2,3}. For all i and j, where i ≠ j, we set $m_{i,j}$=m. The program ms takes this parameter in the form of M=4 Nm (where N=10,000 is the effective population size of each subgroup). We performed simulations for 10,000 different values of M chosen uniformly on the log scale from the range (0.01,100). This is akin to running simulations using 10,000 different values of m in the range ($0.25×10^{-7}$, $0.25×10^{-2}$). For each m, 1000 variants were simulated. Thus, for each m, we calculated the percentage of variants in Yoruba (with allele-count >1) wherein the derived allele was shared with the archaic hominins. The proportion of allele-sharing in simulations equaled or exceeded the proportion (13.7%) observed in real-life at approximately m≤0.0075%.

## Identifying deletions in archaic genomes

The identification of deletions in archaic hominins was predicated on the concept that a deletion in an archaic hominin would correspond to a low read depth in the window of deletion in the hominin's genome. We started with two main types of input files: (1) The VCF file for the 1000 Genomes Phase 3 dataset and (2) a BAM file for each high-coverage archaic hominin.

The 1000G phase-3 VCF file was obtained from https://www.internationalgenome.org/data. This includes 84.4 million variants from 2504 individuals across 26 populations. The file was then converted to a BED file (with three tab-separated columns representing chromosome numbers, start positions, and end positions of deletions) using a script written in AWK. This VCF file was filtered to retain only biallelic autosomal deletions. This amounted to 32,154 deletions. We genotyped all these deletions in the four high-coverage archaic hominin genomes. (Note that in the main text, we focused only on 4863 deletions with both allele-count >1 in YRI, CEU, and CHB combined, and at least one SNV in LD.)

The sequence files for archaic genomes mapped to hg19 were obtained from https://www.eva.mpg.de/genetics/genome-projects.html?Fsize=0%2C%252%27A%3D0. These BAM files containing mapping information (such as the start and end coordinates of the part of the genome to which a read maps) were converted to BED files. This was done using the bamToBed command in the bedtools module (**Quinlan and Hall, 2010**). We then used the two types of BED files to count the number

of reads for each archaic genome that mapped to a region of the genome that is polymorphically deleted in AMHs. In order to achieve this, we used the intersectBed command with the -c option within the bedtools module. This command counts the number of reads in an archaic genome that intersects with the region of the genome harboring a deletion polymorphism in AMHs.

Next, for every archaic genome, we normalized the number of reads at each window of deletion by the size of the window.

$$\text{Normalized Read Depth} \ = \ \frac{\text{\# of reads intersecting the window of deletion}}{\text{Size of the window of deletion}}$$

For each archaic genome, we wanted to calculate the Z-scores of the normalized read depths across all windows of deletion, and classify a window as a deletion if the normalized read depth was below a certain threshold. To prevent outliers from affecting measures of central tendency and spread, and therefore the Z-score threshold, we use the modified Z-score to classify a region as deleted or non-deleted in an archaic genome. The modified Z-score uses median (as opposed to mean) and median absolute deviation (as opposed to standard deviation) to calculate the Z-score. For a given archaic genome, the modified Z-score of the normalized read depth at the ith window of deletion is given by:

$$\text{ModZ}_i \ = \ \frac{r_i \ - \ \text{Median}\,(R)}{\text{MedianAbsoluteDeviation}\,(R)}$$

where $r_i$ denotes the normalized read depth at a given window of deletion, and R denotes the random variable representing normalized read depth.

*Iglewicz and Hoaglin, 1993*, have suggested that a threshold ±3.5 is reasonable for outlier detection using modified Z-scores. Nevertheless, for our purposes, we used a more conservative threshold of –5, which we deemed more appropriate based on spot checking. For example, if the modified Z-score (of the normalized read depth) at a window of deletion was less than –5 in the Vindija Neanderthal, that window was classified as deleted in the Vindija Neanderthal. The distributions of these modified Z-scores across windows of deletions for the four high-coverage archaic genomes are illustrated in *Figure 2—figure supplement 1*. All calculations downstream of obtaining the raw numbers of reads from archaic genomes intersecting with the windows of deletion were performed using a script that we wrote in R. The read depth analysis was done using all 32,154 AHM deletions (results for the status of these deletions in the four high-coverage archaic genomes are available in *Supplementary file 1*).

## Identifying SNVs that are in LD with deletion polymorphisms in AMHs

We subsetted the 1000G phase-3 VCF files (there is a separate file for each chromosome) obtained from https://www.internationalgenome.org/data to retain individuals only from CEU (n=103), CHB (n=99), and YRI (n=108) populations. This filter was applied using the `--keep` option in the module VCFtools (*Danecek et al., 2011*). All variants that had a minor allele-count <2 were eliminated using the `--mac` filter in VCFtools. We used the resulting VCF files to identify SNVs in LD with each of the autosomal biallelic deletions (with minor allele-count >1 in YRI, CEU, and CHB combined) within a 50 kb radius of the deletion. We did this using the `--hap-r2-positions` and `--min-r2` 0.9 flags in VCFtools. For each autosomal biallelic deletion with allele-count >1, this gave us a list of SNVs in LD with the deletion with $r^2 >0.9$, if such SNVs existed. At least one such SNV in LD existed for 4863 deletions. We called this set of deletion the 'deletion dataset' and based all our downstream analysis on it.

It is important to describe why we only focused on deletions with identifiable variants (most of them SNVs) in LD with them. We can only eliminate potentially introgressed deletions by checking whether at least one of the SNVs in LD with a deletion is already known to be introgressed. Moreover, we can confirm whether a deletion shared between archaic hominins and AMHs is identical by descent (thereby eliminating recurrence) by checking whether the same SNVs accompany the deletion in AMHs and archaic hominins. This filtering would not be possible if our deletions were not flanked by variants in LD with them. A shortcoming of this approach is that it fails to capture balanced deletions that are not in LD with at least one SNV.

## Eliminating instances of recurrence and introgression

We found that 575 human polymorphic deletions are also present in at least one archaic hominin genome. In order to ensure that we do further analysis only on deletions that are shared with archaic hominins by common descent, we wanted to eliminate shared deletions that were recurrent or introgressed.

We removed recurrent deletions (those emerging in AMHs and archaic hominins independently) by retaining only those shared deletions for which at least one allele in LD with the deletion in AMHs was also present in at least one archaic genome that harbored the deletion. To do this, we needed to know whether variants in LD with deletions are present or absent in the archaic genomes. We started with two types of inputs: (1) VCF files for each of the archaic genomes (mapped to hg19) and (2) a file containing all the variants (SNVs) in LD with polymorphic deletions in AMHs. We filtered the VCF files to include only the SNVs in LD with shared deletions. This was done using the `--positions` flag in VCFtools. The presence or absence of every variant in LD was then determined using the vcf files for the archaic hominins. The procedure was implemented using an AWK script. 53 shared deletions were classified as 'recurrent' using this approach.

In order to eliminate introgressed shared deletions, we used the results published by *Taskent et al., 2020*. In their study, the authors had identified introgressed haplotypes in Eurasians using the S* statistic. They had also published a list of S*-significant SNVs that characterize introgressed haplotypes. We stamped out the shared deletions that were both absent in Yoruba and for which at least one allele in LD was among the S*-significant variants listed in the study mentioned above. We thus eliminated 92 deletions that were likely introgressed from archaic hominins into AMHs.

## Age of deletions and allele frequency trajectories

We estimate the ages of the deletions in the *deletion dataset* using two methods: (1) Human Genome Dating database (https://human.genome.dating/download/index) and (2) Relate (*Speidel et al., 2019*).

The Human Genome Dating database (https://human.genome.dating/download/index) hosts age estimates for over 45 million SNVs (*Albers and McVean, 2020*). This database reports multiple age estimates for each SNV. We used the median age estimate calculated using the joint clock. Since this database only includes age estimates for SNVs (and not for deletions), we could only date a deletion if the dating database contained the age estimate for at least one of the variants in LD ($r^2 > 0.9$) with the deletion. If age estimates were available for only one variant in LD, the same age estimate was assigned to the deletion. If age estimates were available for more than one variant in LD with the deletion, we used the highest age estimate, which may be inaccurate in certain cases.

Relate is a method that estimates genome-wide genealogies and can be used to infer the age of a variant (*Speidel et al., 2019*). We used Relate to infer the ages of the deletions in the *deletion dataset*. To this end, we used previously inferred genome-wide genealogies for samples of the SGDP dataset (*Mallick et al., 2016*; *Speidel et al., 2021*), available from https://www.dropbox.com/sh/2g-jyxe3kqzh932o/AAAQcipCHnySgEB873t9EQjNa?dl=0. For each deletion, we used SNVs in LD where the derived allele was tagging the deletion at an $r^2$ exceeding 0.9 and calculated the mean age of such SNVs to date each deletion.

To quantify allele frequency variation, we computed the ratio of lineages carrying the derived allele by the total number of lineages remaining at 5000 years and 50,000 years before present, but only if the number of lineages remaining at 50,000 years exceeded 10% of the present-day sample size. We then standardized the allele frequency change stratified by present-day allele frequency, by calculating the mean and standard deviation given present-day frequency. Finally, we squared this standardized allele frequency change to obtain our statistic $\chi^2$, which is expected to have a Chi-squared distribution with one degree of freedom under neutrality, and smaller values for more stable trajectories. This approach was inspired by *Edge and Coop, 2019*, who used a similar approach to quantify polygenic positive selection using genealogies.

## Beta measure

We used a recent and robust measure of balancing selection (*Siewert and Voight, 2017*), stdβ2, to investigate whether ancient deletion are enriched for targets of balancing selection. A high stdβ2 for a variant is indicative of balancing selection.

In our study, we estimated stdβ2 for the deletions in the deletion dataset using SNVs in LD with them. We did this for the CEU, CHB, and YRI population separately. The stdβ2 scores for SNVs are publicly available (https://github.com/ksiewert/BetaScan, copy archived at swh:1:rev:8ade55c-13fa5e21bb858bbe9078e0c67af519a77; *Siewert-Rocks, 2023*) for the CEU, CHB, and YRI populations. For each deletion in each of these populations, we obtained the stdβ2 values for variants in LD with deletions whenever they were available. For a given deletion, when the stdβ2 values were available for more than one variant in LD with the deletion, we used two approaches to estimate the stdβ2 for the deletions. In the first approach, we used the highest stdβ2 among the LD variants as the estimate for the stdβ2 for the deletion. We call this BETAMAX. In the second approach, we focused on the stdβ2 values for the SNVs that were in LD with the deletion with the highest $r^2$ value. If multiple SNVs were in LD with the deletion with the highest $r^2$ value, we used the stdβ2 value of the SNV that was closest to the deletion among these SNVs as the estimate for stdβ2 for the deletion. We call this BETAPRIME. We performed this process for YRI, CEU, and CHB populations separately to arrive at stdβ2 estimates for deletions in our deletion dataset in each of these three populations. Using both BETAMAX and BETAPRIME gave us similar trends across populations. These values are available in *Supplementary file 2*.

## Ascribing phenotypic relevance to deletion

We used two criteria to ascribe phenotypic relevance to deletions: (1) intersection of the deletion with at least one exon and (2) association of an SNV in LD with the deletion with a GWAS trait.

In order to identify deletions that intersect with exons, we started with the genome annotation file download from https://hgdownload-test.gi.ucsc.edu/goldenPath/hg19/bigZips/genes/hg19.refGene.gtf.gz. Using an AWK script, this GTF file was then converted to a BED file containing five columns: (1) annotation's chromosome number; (2) annotation's start position; (3) annotation's end position; (4) gene name; and (5) type of feature. Only the rows wherein the type of feature was 'exon' and the chromosome number was between 1 and 22 were retained. All repeated entries (rows) were eliminated. The resulting file contained only columns (1) to column (4). We then used a BED file containing information about the AMH deletions in our deletion dataset and the BED file mentioned above to identify deletions spanning exons. This was done using the intersectBed option with -wa and -wb flags in the BEDtools module. On the resulting file, we used the groupby tool with the '-o freqdesc' flag in the BEDtools module in order to obtain a file containing the names of the genes (and the number of exons within each intersecting gene) that overlap the deletions; 243 (5%) of the 4863 deletions in the deletion dataset were exonic.

The second method to ascribe phenotypic relevance to deletions was to use results from previously published GWAS. We used a publicly available catalog of GWAS results based on the UK BioBank data (http://www.nealelab.is/uk-biobank/). In particular, we used data for 4113 traits. For each trait, we used data that produced results using both sexes. For continuous traits, we used the raw version of the data, as opposed to the inverse rank normalized version. For each trait, only those SNVs were retained that were associated with the phenotype with a p-value less than $10^{-8}$. We are making available a consolidated table with all statistically significant associations from this dataset (https://figshare.com/articles/dataset/Table_S3_for_Aqil_et_al_2022/19606192). We hope this will make it easier for the scientific community to use GWAS results than the currently available datasets which store associations with each trait in a different table. Then, for each of the 4863 deletions in our deletion dataset, we checked if any of the SNVs in LD were among the SNVs that were significantly ($p<10^{-8}$) associated with a phenotype. We then obtained the phenotype that was associated with one of the SNVs in LD with the lowest p-value, and ascribed it to the deletion. Thus, 433 (8.9%) of the 4863 deletions in the deletion dataset had phenotypic associations.

## Enrichment analysis for ancient deletions

We performed enrichment analyses for phenotypic relevance and length among ancient deletions using variants with a pooled frequency >5% in YRI, CEU, and CHB combined. First, we investigate whether a higher proportion of ancient deletions, relative to non-ancient deletions, have phenotypic relevance. To this end, we defined phenotypic relevance in three ways: (1) GWAS association, (2) exonic overlap, and (3) both GWAS association and exonic overlap. For each definition, we first calculated the observed proportions of phenotypic deletions among both ancient and non-ancient

categories in turn. Then, we shuffled the 'ancient' and 'non-ancient' labels among the deletions in 10,000 permutations, calculating the proportion of phenotypic deletions among both ancient and non-ancient labels for each permutation. Using the number of permutations in which the difference in proportions of phenotypic deletions was more extreme than the difference in observed proportions, we obtained an empirical p-value for phenotypic enrichment among ancient deletions.

We also investigated whether certain phenotypic categories are overrepresented in ancient deletions relative to non-ancient deletions. For this, we used the UK BioBank traits associated with the deletions. In total, 1675 traits were associated with the deletions in the deletion dataset. We manually placed each of these traits into 1 of 18 categories such that any deletion could be associated with one or more phenotypic categories (*Supplementary file 3*). Only deletions with a pooled frequency >5% in YRI, CEU, and CHB combined were retained for analysis. For each phenotypic category, we obtained the proportion of deletions associated with that category among ancient and non-ancient deletions. We then shuffled the 'ancient' and 'non-ancient' labels in 10,000 permutations. Just as above, we used the number of permutations in which the difference in proportions of deletions associated with a phenotypic category was more extreme than the difference in observed proportions, and we obtained an empirical p-value.

Now, we turn to length enrichment. We calculated the 2nd, 5th, 10th, 20th, 30th, 40th, 50th, 60th, 70th, 80th, 90th, 95th, and 98th length percentiles for deletions in ancient and non-ancient categories. We calculated the differences in corresponding percentiles in ancient and non-ancient deletions. Again, we shuffled the 'ancient' and 'non-ancient' labels into 10,000 random permutations, calculating the differences in corresponding percentiles in ancient versus non-ancient deletions for each permutation. This gave us empirical p-values for differences in the length of ancient and non-ancient deletions at various percentiles.

## Simulations to identify signatures of overdominance

We set out to identify signatures associated with a locus that has evolved under overdominance.

Methodologically, we approached the problem of separating overdominance from neutrality on two fronts: (1) the trajectory of the allele frequency of the mutation conditioning on the age of the mutation, and (2) the patterns of neutral polymorphisms around the so-called focal mutation which has evolved either under overdominance or neutrally (with the same age of the mutation). Given that a mutation under overdominance (heterozygote advantage) may be at intermediate frequency, we also (3) studied simple coalescent simulations conditioning on the existence of at least one SNV within a certain frequency range (e.g., 50–60%). The age of a mutation is crucial for the study of balancing selection. We considered two values for the age of a mutation: (1) 10,000 generation and (2) 40,000 generations old. In the first case, the age of the mutation corresponds to 10,000×29 years=290,000 years old mutation. In the second case, the mutation is 1,160,000 years old. Conditioning on the age of the mutation, we generated allele frequency trajectories of the mutation, that is, the frequency of the mutation at each time point from its onset until the present day. For the overdominance scenario, we used the software *trajdemognpops,* implemented using tools from ms and *mssel* (kindly provided by RR Hudson), to generate trajectories of a mutation under overdominance. The dominance coefficient is characterized by a large value (here, h=10) in order to assign a benefit for the heterozygote. Thus, the fitness for genotypes at a biallelic locus is by:

Aa: (1+sh), where s is the selection coefficient and h the dominance coefficient. Here, s=0.005 and h=10, that is, the heterozygote Aa has a fitness value 1.05. If we had considered codominance (as opposed to overdominance), we would have set h=0.5.

AA: 1+s, thus, the fitness for the AA is 1.005 **aa**: 1.

Given the trajectory of a mutation, a population is split into two kinds of genotypes. The haplotypes carrying the mutation (or the derived allele) and the haplotypes that carry the ancestral allele. Each *neutral* (i.e., passenger mutation) can change 'population' (genetic background) by recombination. Therefore, the coalescent in this case is described as a structured coalescent of two populations (a population with the derived allele and another population with the ancestral allele) that communicate between themselves via recombination. The size of each population is determined at each time point by the trajectory of the derived allele.

In order to understand the effect of the age of the allele (and also to test the mssel code for correctness), we performed coalescent simulations (using Hudson's ms) conditioning on the presence

of at least one SNV at frequency within a given range. Since we are interested in the mutations that are approximately at 50% frequency in the population, we conditioned on the presence of derived mutations in the sample in 22–28 (out of 50) haplotypes. This set of simulations are called *pseCoal*. For each simulated dataset, we calculated the relevant population genetics statistics available through Comus (*Papadantonakis et al., 2016*) including number of segregating sites, θ, π, Tajima's D, ZnS, Fay, and Wu's H, dvk, and dvh. Then, we conducted a summary of all these statistics using PCA. The goal is to understand whether the different scenarios can get separated by using polymorphic patterns.

## Conceptual and methodological concerns

### Human polymorphisms wherein derived allele is shared with archaic hominins are older than 700,000 years

AMHs and archaic hominins are estimated to have diverged around ~700,000 years ago. Therefore, if a human polymorphism has been maintained for more than ~700,000 years, it was also present in the common ancestral population of AMH and archaic lineages. It follows that if a polymorphism (the presence of both ancestral and derived alleles in a population) is present in AMHs and archaic hominins, then (barring recurrence and introgression), by parsimony, the polymorphism was also present in their common ancestral population (*Figure 1A*). Thus, a polymorphism that is shared by common descent between AMHs and archaic hominins has been maintained for over ~700,000 years. Moreover, because the ancestral allele is fixed in chimpanzees by definition, AMH polymorphisms wherein the derived allele is fixed in archaic hominins were also present (in the polymorphic state) in the common ancestral population of archaic hominins and AMHs. In essence, AMH polymorphisms for which archaic hominins carry the derived allele (fixed or polymorphic) have been maintained for more than 700,000 years.

Why use SNVs (as opposed to deletion polymorphisms) for comparison of the real-life proportion of ancient polymorphisms against neutrally simulated SNVs.

It is worth explaining why we used SNVs, instead of deletions – the class of variants that we are interested in – for comparing observed versus simulated (under neutral conditions) proportions of ancient polymorphisms. Deletions are not suitable for such a comparison because, in general, they are targets of strong negative selection (*Conrad et al., 2006*; *Kondrashov, 2017*; *Lin et al., 2015*; *Lin and Gokcumen, 2019*). Thus, negative selection would have purged a large proportion of deletions that emerged in the common ancestral population of AMHs and archaic hominins. It follows that a smaller proportion of AMH polymorphic deletions than expected under neutral conditions will be shared by common descent with archaic hominins. Even if balancing selection were inflating the proportion of deletions that are shared with archaic hominins, it would not be observable due to the opposite deflationary effect of negative selection. Since negative selection is not as strong a force in the evolution of SNVs as it is for deletions, this problem would not be as pronounced if we used SNVs instead of deletions for testing this premise. Hence, our choice of SNVs for this analysis. In the main text, we have expanded on the idea that deletions are targets of stronger negative selection.

### The vast majority of human deletions are derived relative to chimpanzees

In order to identify deletion polymorphisms that have been maintained in the human lineage for over 700,000 years, we focused on deletions that were present (either polymorphically or fixed) in the four high-coverage archaic genomes. This technique would work only for deletions that were derived in humans, relative to chimps. However, variants that have been called as deletions (wherein deletion is the alternative allele) in the 1000 Genomes Project may, in fact, be human-specific insertions, such that the reference allele (non-deletion) is derived. To investigate how common this situation is among the 4863 deletions in our dataset, we lifted over the the coordinates of the deletions from hg19 on to the Chimpanzee reference panTro3 using the LiftOver tool in UCSC Genome Browser (*Kent et al., 2002*). If the liftover for a deletion fails on account of the window being completely or partially deleted in the Chimpanzee reference, it is indicative of the region being a human-specific insertion. The liftover failed for this reason for only 184 (3.8%) of the 4863 deletions. Therefore, the vast majority of deletions are, in fact, derived relative to chimps.

## Data availability

All data generated can be found in the supplementary files. The codes that we used to generate our datasets and simulations can be found either in Methods or on our GitHub page (https://github.com/GokcumenLab). The consolidated file including all significant UK BioBank GWAS associations for SNVs is available at our FigShare (https://figshare.com/articles/dataset/Table_S3_for_Aqil_et_al_2022/19606192).

## Acknowledgements

We thank Dr Victor Albert and Dr Vincent Lynch for their careful reading of this manuscript. We acknowledge Petar Pajic and Charikleia Karageorgiou for their insightful discussions throughout the development of this project. FUNDING: OG acknowledges support from the National Science Foundation (Grant No. 2123284). LS is funded by a Sir Henry Wellcome fellowship (220457/Z/20/Z). This research was funded in part by the Wellcome Trust. For the purpose of Open Access, the authors have applied a CC BY public copyright license to any Author Accepted Manuscript version arising from this submission.

## Additional information

### Funding

| Funder | Grant reference number | Author |
| --- | --- | --- |
| National Science Foundation | 2123284 | Omer Gokcumen |
| Sir Henry Wellcome Fellowship | 220457/Z/20/Z | Leo Speidel |
| Wellcome Trust | | Leo Speidel |

The funders had no role in study design, data collection and interpretation, or the decision to submit the work for publication. For the purpose of Open Access, the authors have applied a CC BY public copyright license to any Author Accepted Manuscript version arising from this submission.

### Author contributions

Alber Aqil, Conceptualization, Data curation, Formal analysis, Visualization, Methodology, Writing - original draft, Writing - review and editing; Leo Speidel, Pavlos Pavlidis, Conceptualization, Formal analysis, Methodology, Writing - review and editing; Omer Gokcumen, Conceptualization, Resources, Formal analysis, Supervision, Funding acquisition, Visualization, Methodology, Writing - original draft, Project administration, Writing - review and editing

### Author ORCIDs

Alber Aqil ⓘ http://orcid.org/0000-0002-6784-6495
Leo Speidel ⓘ http://orcid.org/0000-0002-4644-8033
Pavlos Pavlidis ⓘ http://orcid.org/0000-0002-8359-7257
Omer Gokcumen ⓘ http://orcid.org/0000-0003-4371-679X

### Ethics

This study investigated variation in previously published anonymized genome data from the 1000 Genomes Project.

### Decision letter and Author response

Decision letter https://doi.org/10.7554/eLife.79111.sa1
Author response https://doi.org/10.7554/eLife.79111.sa2

## Additional files

### Supplementary files
• Supplementary file 1. This table contains the status, in the 4 high-coverage archaic hominin genomes, of all 32,152 biallelic deletions reported in the 1000G Phase-3 dataset.
• Supplementary file 2. This table contains information about the 4,863 deletions that we used in our analysis.
• Supplementary file 3. This table categorizes, into 18 categories, the 1,675 traits associated with the 4,863 deletions.
• MDAR checklist

### Data availability
All data that are used in the study can be found publically. The references and databases are provided in the manuscript. The code and resulting datasets are all provided either through our laboratory's GitHub page, FigShare, or as supplementary tables.

The following dataset was generated:

| Author(s) | Year | Dataset title | Dataset URL | Database and Identifier |
|---|---|---|---|---|
| Aqil A | 2022 | Curated UK Biobank traits | https://figshare.com/articles/dataset/Table_S3_for_Aqil_et_al_2022/19606192 | FigShare, 19606192 |

The following previously published datasets were used:

| Author(s) | Year | Dataset title | Dataset URL | Database and Identifier |
|---|---|---|---|---|
| Sudmant et al | 2015 | An integrated map of structural variation in 2,504 human genomes | https://www.nature.com/articles/nature15394 | Database of Genomic variants: estd219, estd219 |
| Neale et al | 2018 | UK Biobank - Curated | http://www.nealelab.is/uk-biobank,https://docs.google.com/spreadsheets/d/1kvPoupSzsSFBNSztMzl04xMoSC3Kcx3CrjVf4yBmESU/edit#gid=227859291 | biobank, 227859291 |
| Londsdale et al | 2013 | GTEX | https://www.nature.com/articles/ng.2653?report=reader | dbGaP accession number phs000424.vN.pN, phs000424.vN.pN |

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
