## [Editor Report]

Detecting and quantifying balancing selection is a notoriously difficult challenge. In this study, the authors use both empirical analyses and simulations to characterize the amount of balancing selection in the human genome, focusing specifically on the contribution of polymorphic deletions. These results will be of interest to population and human geneticists.

---

## [Decision Letter]

**Decision letter after peer review:**

Thank you for submitting your article "Balancing selection on genomic deletion polymorphisms in humans" for consideration by *eLife*. Your article has been reviewed by 2 peer reviewers, and the evaluation has been overseen by a Reviewing Editor and Detlef Weigel as the Senior Editor. The reviewers have opted to remain anonymous.

Essential revisions:

Both reviewers agree that this paper is interesting and should warrant publication in *eLife*, assuming that the authors can perform the following essential revisions, which are laid out in more detail in the individual reviews:

1. Show that the results a robust to variation of the neutral demographic model used. In particular, the departure presented in Figure 1B might depend critically on the specific population sizes chosen both in modern and in archaic humans. The authors should repeat their simulations while varying these population sizes.

2. Please provide some estimation of the fraction of deletions that are truly novel balancing selection loci as opposed to simply being linked to balancing selection loci already identified through linked SNVs in the cited scans.

3. Please provide some more justification/assessment for the classification of the shared polymorphisms as recurrent, recently introgressed, or ancient shared by descent and the use of SNPs with r2 > 0.9.

4. If the authors can show that there is statistical enrichment of functional deletions among the ancient set, that would strengthen the conclusion and their "27%" statement in the abstract.

*Reviewer #1 (Recommendations for the authors):*

The reasons why the authors use a random subset of variants to measure allele sharing instead of using all SNV variants available are very important and needs to be extensively explained in the main text. Is it to avoid the confounding effect of linkage blocs?

The authors already have the Relate allele age estimates. In addition to allele sharing, the authors could also compare the distribution of ages of shared alleles under the neutral simulations with those observed in the genome.

It would be great to have Figure 1B and its distributions of allele sharing, but multiple times, each time for a separate bin of derived allele frequencies, for example 0-0.05, 0.05-0.1, 0.1-0.2, 0.2-0.3,…..0.8-0.9, 0.9-0.95, 0.95-1. Even if the authors mention that simple overdominance does not seem to explain the results, it is important to be able to check that the excess of shared alleles is not concentrated on low minor allele frequencies.

The use of human-specific as in Figure 2 is erroneous. Every Homo is human. The authors should use modern human-specific.

In figure 4 the x-axis should be labeled stbeta2, not just β.

*Reviewer #2 (Recommendations for the authors):*

1. Regarding the classification of the shared variants as recurrent, ancient shared by descent, and recently introgressed:

– In the SNP analysis, were CpG polymorphisms included? Usually these need to be carefully considered as they are more likely to be recurrent mutations. Is this proportion still elevated among the ancient set, and is there still an excess of ancient polymorphisms if CpG sites are excluded from both dataset and simulations?

– Perhaps the first issue, and eliminating recurrent events more generally, is resolved by using the LD criteria. However, this is not directly addressed by the authors but rather assumed to be the case. The polymorphisms are considered ancient when there is the same SNP in high LD (r2 > 0.9) in both modern human and archaic genomes. How good is this at differentiating ancient from recurrent cases? I.e., how often do you expect to have a SNP in this LD in both populations if recurrent? How often do you expect not to have a SNP in LD if shared by descent? Why was 0.9 chosen? If the deletions are truly shared by descent, they should have the same breakpoints – was this always the case to the resolution that could be assessed?

2. The selection of variants by looking only at YRI, CEU, and CHB seems odd, both because CHB has ~ half the sample size of the other populations and it seems that any/all populations could be used. If there is a reason for this, it would be good to include it.

3. The authors suggest that requiring a SNP to be in high LD (r2>0.9) with the deletions does not introduce any bias in the proportion shared by common descent. One concern could be that this would bias towards regions of low recombination, although I'm not sure what effect that would have on either proportion shared by common descent or recurrent mutation, though the effects of background selection might be greater. The authors could test whether there is a bias to regions of low recombination and if so how that could affect the analysis. Is there any effect on the length of the deletions (and more generally, what is the distribution of lengths of these deletions in the three classes)?

4. Along these same lines, it is not stated in the main text (unless I missed it) but the detail is relevant that the ages are for the SNP in LD, not the deletion itself. It should be clearer that this is being used as a proxy (not just in methods) and also some indication of how close these age estimates are expected to be with r2=0.9.

Writing comments

5. An explanation of what "sharing with archaics" means was lacking in the main text – it is only late in the methods that it says the four genomes are 3 Neandertals and 1 Denisovan genome, and whether this sharing had to be with all or any of them, heterozygous or homozygous, to be considered as shared. Given how key this is to understanding what the set of variants analysed consists of, I think it needs to be briefly stated when the dataset is introduced in the results.

6. The methods are also in a very different order than the results are presented (e.g., simulations for the non-synonymous variants are first in the results but late in the methods, association with biobank phenotypes is early in the methods but late in the results). It would help the reader and the flow if these could be more aligned (I realize that is not always strictly sensible).

7. Sometimes "linkage" or "linked" is used when it should be "linkage disequilibrium", e.g., "we imposed this linkage requirement" and in the table README, but LD is what is meant.

8. Happened to notice typo in last line of p. 18 ('classified as phenotypically insignificant').

---

## [Author Response]

Essential revisions:Both reviewers agree that this paper is interesting and should warrant publication in eLife, assuming that the authors can perform the following essential revisions, which are laid out in more detail in the individual reviews:1. Show that the results a robust to variation of the neutral demographic model used. In particular, the departure presented in Figure 1B might depend critically on the specific population sizes chosen both in modern and in archaic humans. The authors should repeat their simulations while varying these population sizes.

We agree with the reviewers that this is a crucial point. To address this as comprehensively as possible, we have now generated additional simulations with varying parameters, as suggested by the reviewers. We found no realistic neutral scenarios that explain the excess allele sharing that we observed. We now reconstructed Figure 1 based on these findings and described them as follows in the Results section:

“To ensure that our analysis is not biased by the idiosyncrasies of any particular model, we performed these simulations using 36 distinct models. The models vary by three parameters: N_e_ of Yoruba/AMH, N_e_ of archaic hominins, and the time of divergence between the AMH and the archaic hominin lineage. The N_e_ for humans can be either constant (ranging from 10,000 to 30,000) or varying over time (Speidel et al. 2019) (Figure 1B). N_e_ of archaic hominins ranges from 1,000 to 3,000; and the divergence time ranges from 500 to 700 kya (Bergström et al., 2021; Mafessoni et al., 2020; Meyer et al., 2012; Prüfer et al., 2014). In the main text, we focus on the model with variable AMH-N_e_ (Speidel et al. 2019), using the well-accepted archaic hominin N_e_ of 1,000 and a divergence time of 700 kya. We refer to this as the “base model”.

Using this model, we find that the entire simulated distribution of the proportion of ancient polymorphisms lies to the left of the empirical value of 13.7% (Figure 1C). These results hold for all other models with realistic sets of parameters. Even when we change any single parameter to unrealistic levels (e.g., either AMH-N_e_ = 30,000, or divergence time = 500 kya), we still observe an excess of ancient polymorphisms. Neutral models can explain the empirical proportion of ancient polymorphisms only when at least two parameters are tweaked in a less realistic direction (e.g., both AMH-N_e_ = 30,000 and divergence time = 500 kya). Therefore, we conclude that the high proportion (13.7%) of ancient polymorphisms cannot be explained by realistic neutral scenarios.”

2. Please provide some estimation of the fraction of deletions that are truly novel balancing selection loci as opposed to simply being linked to balancing selection loci already identified through linked SNVs in the cited scans.

Based on this suggestion, we have compared the haplotypes that harbor ancient deletions and previous balancing selection datasets and found that virtually all of our candidate deletions are novel. We described this analysis in a paragraph:

**“**Previous genome-wide balancing selection scans focused on either individual genes or single nucleotide variants (SNVs). Consequently, we do not expect to find many ancient deletions that have previously been reported as targets of balancing selection. Nevertheless, we investigated whether the exons in any of the genes that have been reported as targets of balancing selection in DeGiorgio et al. (2014) or Soni et al. (2022) overlap with ancient deletions. We found no overlaps. We also found that 77 common (>5%) ancient deletions were linked (r2>0.9) to SNVs that had high (in the 95th percentile) Std*β*2 (Siewert and Voight 2020) values in YRI, CEU, and CHB. This is unsurprising since this is the measure we used to show that ancient deletion polymorphisms are enriched for balancing selection targets. Interestingly, one of the ancient deletions with a high associated Std*β*2 value overlaps a candidate region for balancing selection previously identified using the non-central deviation (*NCD*) method (Bitarello et al. 2018). This 433 bp deletion (esv3607090), which is 2 million years old and common across populations, deletes part of an intron of the *STK32A* gene. This could be an interesting subject for future studies. Regardless, a vast majority of common ancient deletions (73%) were not reported previously as balancing selection candidates and thus are novel targets for future studies.”

3. Please provide some more justification/assessment for the classification of the shared polymorphisms as recurrent, recently introgressed, or ancient shared by descent and the use of SNPs with r2 > 0.9.

We used r^2^ = 0.9 as an arbitrary but conservative threshold regarding the recurrent variants. Specifically, the higher the r^2^ threshold, the lower the chances of us categorizing a recurrent variant as an ancient one. Given that this is a crucial issue and a major point raised by the reviewers, we conducted additional analyses considering CpG sites and allele frequency spectra to ensure that recurrent mutations do not bias our results. We described these new analyses in the Results section as follows:

“There is a remote possibility that we are observing an empirical excess of ancient polymorphisms owing to a high incidence of recurrent mutations in the AMH and archaic hominin lineages that remained undetected by the linkage-disequilibrium-based approach we used to identify them. Since CpG sites may be particularly prone to recurrent mutations, we calculated the proportion of empirical ancient polymorphisms again using only A_⇄_T SNVs. This analysis yielded a proportion of 14.24%, not much different from the previously calculated 13.7%. Moreover, if recurrence was the main cause of the observed excess of ancient polymorphisms, we would expect this excess (relative to the simulated distribution) to be more pronounced among polymorphisms with low derived allele frequencies. To test if this is the case, we repeated our analysis using the base model for simulations, dividing the empirical and simulated SNVs into derived allele frequency bins (Figure S1). We observed that the excess of ancient polymorphisms is, in fact, most pronounced at high derived allele frequency. Both these results combined suggest that our results are not biased due to undetected recurrent mutations.”

4. If the authors can show that there is statistical enrichment of functional deletions among the ancient set, that would strengthen the conclusion and their "27%" statement in the abstract.

We thank the reviewers for this comment (and the related comments in individual reviews) because it helped us to acquire novel and meaningful results, improving our understanding of these deletions substantially. We described these findings in a new section and figures. We added a new section describing these results. It reads:

“We did not observe a significant increase in either the proportion of exonic (ancient=5.6%; non-ancient=3.6%) or GWAS-associated (ancient=19.1%; non-ancient=16.4%) ancient deletion, relative to non-ancient deletions (Figure 5B). However, when we classify a deletion as functional more conservatively, *i.e.,* it both intersects an exon and has a GWAS association (*i.e.,* one of the SNVs linked to it has a GWAS association), we observe a 4.2-fold enrichment (p=0.003) of functional variants among ancient deletions (Figure 5B). In fact, out of the 18 common deletions (frequency>5%) that both intersect genes and have GWAS associations, 9 (50%) are ancient. Further, we observed a phenotypic enrichment among ancient deletions for some GWAS trait categories: a 12.5-fold enrichment (p=10^-5^) of traits related to inflammatory response and a 2.8-fold enrichment (p=0.003) of traits related to metabolism (Figure 5C).”

Reviewer #1 (Recommendations for the authors):The reasons why the authors use a random subset of variants to measure allele sharing instead of using all SNV variants available are very important and needs to be extensively explained in the main text. Is it to avoid the confounding effect of linkage blocs?

Yes. The reviewer is completely right that we tried to avoid linkage blocks. We now explicitly state that in the main text as follows:

“We focused on random SNVs instead of all the SNVs in order to mitigate the biases that would be introduced due to linkage.”

The authors already have the Relate allele age estimates. In addition to allele sharing, the authors could also compare the distribution of ages of shared alleles under the neutral simulations with those observed in the genome.

This is a good suggestion. However, since our simulation framework for this study is ms-based, where we essentially simulated allele frequencies across different lineages. Thus, we were not able to use haplotype-based measures of age. Having said that, as discussed in the next response, we conducted a more thorough analysis of allele frequency distributions, which are correlated with the age of the allele, between simulated and empirical data.

It would be great to have Figure 1B and its distributions of allele sharing, but multiple times, each time for a separate bin of derived allele frequencies, for example 0-0.05, 0.05-0.1, 0.1-0.2, 0.2-0.3,…..0.8-0.9, 0.9-0.95, 0.95-1. Even if the authors mention that simple overdominance does not seem to explain the results, it is important to be able to check that the excess of shared alleles is not concentrated on low minor allele frequencies.

Thanks for this suggestion. We conducted this analysis, and concordant with our expectations, we observed that the excess allele sharing is much stronger among common alleles as compared to rare alleles. We also did not observe an enrichment of sharing among intermediate allele frequency alleles. We now described these observations in the Results section and a new Figure S1.

The use of human-specific as in Figure 2 is erroneous. Every Homo is human. The authors should use modern human-specific.

We now changed this figure (and others throughout the manuscript) based on the reviewer’s suggestion.

In figure 4 the x-axis should be labeled stbeta2, not just β.

We now changed this figure based on the reviewer’s suggestion.

Reviewer #2 (Recommendations for the authors):1. Regarding the classification of the shared variants as recurrent, ancient shared by descent, and recently introgressed:– In the SNP analysis, were CpG polymorphisms included? Usually these need to be carefully considered as they are more likely to be recurrent mutations. Is this proportion still elevated among the ancient set, and is there still an excess of ancient polymorphisms if CpG sites are excluded from both dataset and simulations?

This is a good point. We now tested this and added this to our Results:

“Since CpG sites may be particularly prone to recurrent mutations, we calculated the proportion of empirical ancient polymorphisms again using only A_⇄_T SNVs. This analysis yielded a proportion of 14.24%, not much different from the previously calculated 13.7%.”

– Perhaps the first issue, and eliminating recurrent events more generally, is resolved by using the LD criteria. However, this is not directly addressed by the authors but rather assumed to be the case. The polymorphisms are considered ancient when there is the same SNP in high LD (r2 > 0.9) in both modern human and archaic genomes. How good is this at differentiating ancient from recurrent cases? I.e., how often do you expect to have a SNP in this LD in both populations if recurrent? How often do you expect not to have a SNP in LD if shared by descent? Why was 0.9 chosen? If the deletions are truly shared by descent, they should have the same breakpoints – was this always the case to the resolution that could be assessed?

We agree with the reviewer that these are important issues. To address these concerns, we conducted additional analysis on recurrence as described in the Results section (please refer to our response to the general reviews). Briefly, we expect recurrent mutations to have lower allele frequencies. We found an opposing trend among ancient deletions (Figure S1). We also repeated our pipeline while ignoring the CpG sites, which are more prone to recurrence, and found similar results. In addition, we conducted our analysis with perfect LD (R^2^=1) and showed that our results remain nearly identical (results not shown). We refrained from breakpoint analysis, even though it may be informative because the breakpoint resolution is very poor in the fragmented short reads of ancient genomes. Overall, these additional analyses boosted our confidence that our pipeline to filter recurrent deletions are robust.

2. The selection of variants by looking only at YRI, CEU, and CHB seems odd, both because CHB has ~ half the sample size of the other populations and it seems that any/all populations could be used. If there is a reason for this, it would be good to include it.

The reviewer is right that we could have used other populations. However, the populations that we selected have been historically well described, and detailed parameters for their demographic histories are available. The CHB (n=103) has a similar sample size to YRI (n=108) and CEU (n=99) in our analysis. We now added the sample size from these populations in the Methods to clarify this point.

3. The authors suggest that requiring a SNP to be in high LD (r2>0.9) with the deletions does not introduce any bias in the proportion shared by common descent. One concern could be that this would bias towards regions of low recombination, although I'm not sure what effect that would have on either proportion shared by common descent or recurrent mutation, though the effects of background selection might be greater. The authors could test whether there is a bias to regions of low recombination and if so how that could affect the analysis. Is there any effect on the length of the deletions (and more generally, what is the distribution of lengths of these deletions in the three classes)?

We thank the reviewer for this comment. We now considered recombination more carefully and added the following to our Methods section:

“There seems to be one bias that may be introduced by eliminating polymorphisms with no SNVs in LD in AMHs: this would bias our analysis to regions with low recombination rates. In these regions, we would expect higher background selection due to the Hill-Robertson effect, leading to a deflation in the proportion of ancient polymorphisms. Since we observe a larger than expected proportion of ancient polymorphisms despite this bias, we can conclude that this bias only makes our analysis more conservative.”

We thank the reviewer for the suggestion on size distribution analysis, which now leads to novel insights. Specifically, we now described our results as follows in the manuscript:

“…we expect that larger deletions should, on average, experience more intense negative selection. Since most large ancient deletions would have been purged by negative selection, we expect surviving ancient deletions to be, on average, smaller than non-ancient deletions. We test this using common deletions (frequency > 5% in YRI, CEU, and CHB combined). Ancient deletions are indeed 14% shorter than non-ancient deletions (p=0.02; permutation test). Nevertheless, there is an excess of long deletions among ancient deletions relative to non-ancient deletions (Figure 7A). In particular, the 95th and 98th size percentiles of ancient deletions are 33% (p=0.04; permutation test) and 128% (p=0.005; permutation test) larger than non-ancient deletions, respectively (see Methods). This excess of longer deletions is inconsistent with evolution under neutrality or negative selection. Therefore, the longest 5% of ancient deletions are excellent targets for future studies of balancing selection. In fact, 3 out of the 9 GWAS-associated common exonic deletions intersecting *SIRPB1*, *LCE3A*, *LCE3B*, and *UGT2B28* are in the 95th percentile of the size distribution of ancient deletions.”

4. Along these same lines, it is not stated in the main text (unless I missed it) but the detail is relevant that the ages are for the SNP in LD, not the deletion itself. It should be clearer that this is being used as a proxy (not just in methods) and also some indication of how close these age estimates are expected to be with r2=0.9.

The reviewer is right. We now state this in the Results sections as follows:

“To confirm that our pipeline for identifying ancient deletions has high accuracy, we estimated the ages of the deletions, based on the ages of SNVs in LD.”

Writing comments5. An explanation of what "sharing with archaics" means was lacking in the main text – it is only late in the methods that it says the four genomes are 3 Neandertals and 1 Denisovan genome, and whether this sharing had to be with all or any of them, heterozygous or homozygous, to be considered as shared. Given how key this is to understanding what the set of variants analysed consists of, I think it needs to be briefly stated when the dataset is introduced in the results.

We now clearly stated this early in the Results section as follows: “A variant was classified as ancient if the derived allele was shared, by common descent, with at least one of the four high-coverage archaic hominin genomes (three Neanderthals and one Denisovan) (Mafessoni et al., 2020; Meyer et al., 2012; Prüfer et al., 2014, 2017)

6. The methods are also in a very different order than the results are presented (e.g., simulations for the non-synonymous variants are first in the results but late in the methods, association with biobank phenotypes is early in the methods but late in the results). It would help the reader and the flow if these could be more aligned (I realize that is not always strictly sensible).

We now reorganized our Methods section as per the reviewer’s suggestion.

7. Sometimes "linkage" or "linked" is used when it should be "linkage disequilibrium", e.g., "we imposed this linkage requirement" and in the table README, but LD is what is meant.

We apologize for this. We now use these terms correctly to avoid any confusion.

8. Happened to notice typo in last line of p. 18 ('classified as phenotypically insignificant').

Thanks for noticing this. We now revised the whole paragraph to clarify our classification pipeline and eliminate the use of phenotypically insignificant.